# Palliative care follow-up for cancer patients combining day-hospital visits and telemedicine: What feasibility?

Valerie Mauries-Saffon[1], Alfonsina Faya-Robles[2,3], Marie Bourgouin[1], Sebastien Lamy[4,5], Nathalie Caunes-Hilary[1], Bettina Couderc[1,3]*

1 Department of palliative care, Claudius Regaud-Oncopole, Toulouse cedex 9, France, 2 Regard Social (Association loi 1901 - Sciences Humaines et Sociales, Santé, sociétés), Toulouse, France, 3 Department Bioethics, CERPOP, UMR 1295, Inserm – Toulouse University: "Trajectoires d'innovations en santé: enjeux bioéthiques et sociétaux" Toulouse cedex, France, 4 Department Equity, CERPOP, UMR 1295, Inserm - Toulouse University: "Trajectoires d'innovations en santé: enjeux bioéthiques et sociétaux", Toulouse cedex, France, 5 Unit for prevention and reduction in cancer inequalities, Claudius Regaud Oncopole, Toulouse cedex 9, France

* Bettina.couderc@inserm.fr

## Abstract

### Background

Due to the increasing life expectancy of patients with cancer, palliative care services are increasingly solicited to provide follow-up for individuals living at home. The implementation of teleconsultations represents a potential solution to extend access to palliative care follow-up to a larger number of patients.

The aim of this study is to demonstrate the a priori acceptance of teleconsultations among all patients admitted to palliative care, as well as the feasibility of their implementation. We therefore initially explored whether a specific patient profile might limit acceptance or feasibility of teleconsultations (e.g., older patients, patients unfamiliar with digital technologies, gender differences, social status, economic status, or place of residence), and we identified patients' expectations and concerns regarding teleconsultation. The secondary objective is to describe the contribution of monthly remote consultations (telemedicine), conducted in parallel with quarterly day-hospital follow-up by a palliative care team, for patients with cancer, in anticipating potential deterioration in their overall clinical condition.

### Methods

We included 36 patients at the time of their first admission to the palliative care day hospital. Upon enrollment, participants completed a questionnaire on demographic characteristics as well as their expectations and/or reluctance regarding teleconsultations. Patients were scheduled to receive monthly teleconsultations (at 1 and 2 months), followed by an in-person visit at 3 months.

**Data availability statement:** All relevant data are within the paper and its Supporting information files.

**Funding:** The author(s) received no specific funding for this work.

**Competing interests:** No authors have competing interests.

## Results

With regard to feasibility and acceptability, our findings indicate that age, sex, socio-economic status, or underlying disease did not influence patient adherence or the successful conduct of teleconsultations. Patients expressed few negative preconceptions regarding their inclusion in the study. Concerning the clinical contribution of monthly consultations, interviewed patients reported a perceived benefit of monthly teleconsultations for both their psychological support and clinical management, and expressed a desire to continue monthly follow-up by teleconsultation beyond the three-month study period.

## Conclusions

In the absence of the possibility of monthly in-person consultations, we highlight the relevance of developing teleconsultations for patients receiving palliative oncological care, regardless of their profile.

---

## Introduction

Life expectancy for patients with cancer has steadily increased in recent years, largely due to advances in treatment and supportive care (including therapeutic innovations, targeted therapies, immunotherapies) as well as screening programs enabling earlier diagnoses [1,2]. The likelihood of being cured of cancer varies according to tumor location, histological type, stage at diagnosis, country of residence, and age. Currently, between 21–47% of men and 38–59% of women can expect to be cured of their cancer [3]. While encouraging, these figures also indicate that approximately 50% of patients will either present with an immediately poor prognosis or progress to a chronic cancer trajectory lasting from several months to years.

Patients with chronic cancer are typically followed at regular intervals by their oncologist before being referred to a palliative care team, who assumes ongoing management [4–6]. For patients with incurable disease, palliative care specialists provide tailored interventions not only aimed at symptom control (e.g., slowing disease progression in collaboration with oncologists) but also at ensuring quality of life (pain and anxiety management, among others) [7,8]. The number of patients requiring palliative care is increasing annually, and the shortage of dedicated palliative care beds has become increasingly apparent in recent years [9–11].

This lack of capacity for day-hospital and inpatient palliative care has prompted policymakers to explore strategies to improve access to supportive and palliative services. Solutions under consideration include the creation of additional beds, new palliative care teams, and alternative care models. Teleconsultation, defined as the use of telecommunication technologies to provide secure medical consultations remotely between a patient and their physician (via computer, tablet, or smartphone), has emerged as a potential tool to improve healthcare delivery. Teleconsultations were first implemented in France in 2009, with reimbursement introduced in 2018 [12].

Although telemedicine can never fully replace in-person visits [13], it may complement existing care by providing timely access to medical advice while awaiting an in-person appointment. In palliative care specifically, teleconsultations could enhance symptom management and improve patients' quality of life through continuous monitoring and timely interventions. They may also reduce emergency hospitalizations between scheduled in-person visits.

Prior implementations of teleconsultations in palliative care (primarily outside oncology or in general practice) have shown high satisfaction among patients, caregivers, home healthcare providers, and clinicians [14]. Several observations underlie this positive perception:

- A randomized clinical trial demonstrated that telemedicine facilitates direct, patient-centered communication between palliative care physicians, patients, caregivers, and home providers, thereby improving collaboration and care integration [14].

- Other studies have highlighted the value of involving a home care nurse during teleconsultations to promptly communicate updates to the primary care physician [15–17]. Improved communication between hospitals and home-care providers reduces caregiver burden; caregivers are relieved not only from the logistical effort of transporting patients to distant appointments but also from the mental load of updating home-care teams after each hospital visit [18].

- Similarly, monthly teleconsultations can alleviate family burden by providing accessible support and reducing the need for emergency care [19].

Teleconsultation thus promotes better integration of primary and specialized palliative care, enhancing continuity of care and treatment planning. With respect to quality of care, multiple studies show that telemedicine strengthens coordination between home-based services and hospital specialists, ensuring that patients receive comprehensive care tailored to their needs [16]. While some studies suggest that frequent teleconsultations may increase symptom reporting due to heightened attention to distress [20], most indicate that continuous monitoring with timely interventions improves symptom management and quality of life in palliative patients [18,19,21].

For clinicians, telemedicine offers time savings, shorter appointment wait times, and reduced no-show rates, making it an efficient method of delivering palliative care. It also helps limit unnecessary hospital admissions and transfers, thereby reducing overall healthcare system burden [19,21]. Importantly, teleconsultations may mitigate health inequities by extending care access to a broader patient population. However, this potential is tempered by challenges related to the cultural and social context of care delivery, requiring careful consideration of the roles and responsibilities of legislators, policymakers, and clinicians [15]. Beyond geographical barriers, socio-economic factors (e.g., lack of digital equipment, internet access, or adequate home environments) may also hinder access to telemedicine.

The present qualitative study aimed to define the parameters for implementing monthly teleconsultation follow-up for patients with chronic cancer referred for the first time to the supportive care department of a comprehensive cancer center. Due to limited staff and bed capacity, most centers can currently offer only quarterly palliative care follow-up visits. However, many patients experience rapid health deterioration between these 3-month in-person appointments, leading to emergency hospitalizations or intensive care admissions. Such outcomes are inconsistent with end-of-life care goals, which emphasize quality of life and discourage aggressive interventions during the last month of life. The proposal of monthly teleconsultations could enable improved management of sudden deteriorations in health status and the associated distressing symptoms.

Before establishing routine monthly teleconsultations in oncology, we sought to evaluate their feasibility and acceptability in an unselected group of patients. Specifically, we aimed to determine whether teleconsultations are suitable for all patients or if they disproportionately benefit specific demographic or socio-economic groups (which would raise concerns about equitable care). We also examined the appropriateness of teleconsultations in light of the unique aspects of supportive care in oncology.

## Materials and methods

### Patient recruitment and follow-up

We recruited patients for this study from individuals attending their first consultation in the palliative care day-hospital setting, who had not previously met the physicians of the mobile palliative care team and who expressed a desire for regular follow-up. Standard follow-up consists of quarterly day-hospital (DH) visits for oncology palliative care. During these visits, the palliative care team conducts a comprehensive assessment of the patient (physical and psychosocial), coordinates hospital–home transitions, arranges home-based support, and facilitates interdisciplinary discussions regarding future oncologic and symptomatic care. Supportive care services to be provided at home (e.g., occupational therapy, sophrology, psychological support) are also organized during these visits.

Inclusion criteria were: adult cancer patients referred for the first time to the Mobile Palliative Care Team (MPCT), owning a smartphone or computer/tablet with internet access, and having an estimated life expectancy of ≥3 months. This latter criterion was based on clinical parameters (absence of major involvement of vital organs: hepatic, pulmonary, or cerebral), a WHO performance status ≤ 3, and the clinical judgment of the pathology-specific oncologist referring the patient to the palliative care department, who estimated a life expectancy greater than three months. The protocol was offered to patients consecutively, in order of admission, without exclusion criteria (except for the absence of connected devices). Acceptability was defined as the proportion of patients who agreed to participate in teleconsultations in addition to their usual follow-up care, relative to the total number of patients to whom this follow-up was proposed. Patients were given an information sheet at the beginning of the day. At the end of the day, the investigator addressed their questions and obtained written informed consent.

At inclusion, patients were informed that participation did not prevent them from contacting the care team at any time or attending the hospital in case of clinical deterioration.

A follow-up schedule was defined for each participant: two monthly teleconsultations (at 1 and 2 months) and an in-person reassessment at the DH at month 3. In all cases, the physician who conducted the initial DH consultation also conducted subsequent follow-up visits.

### Teleconsultations

Patients received an email reminder with the videoconference link three days prior to each teleconsultation. We assessed the feasibility of teleconsultations by calculating the number of patients who received an appointment invitation and who effectively connected to the teleconsultation three days later. On the scheduled day, the MPCT physician who had previously met the patient connected online to conduct a comprehensive assessment similar to the DH visit. The teleconsultation included evaluation of distressing symptoms, functional autonomy, nutritional status, and psychological state. The presence or absence of home support, including a primary caregiver, and any social issues were systematically reviewed. Often, the patient connected together with the primary caregiver, allowing for a more comprehensive assessment.

Video enabled the physician to assess the patient's and caregiver's general condition (pain, weight loss, asthenia, functional status) and visualize symptom locations (e.g., pain sites). Life projects already completed or planned by the patient were also discussed. At the end of the teleconsultation, the physician summarized the findings and, if needed, issued a prescription (sent by email to the patient or directly to the pharmacy). The date of the next teleconsultation or DH reassessment was communicated in the same time, and a care coordination update was shared with the patient's primary care physician or home-care structure (e.g., Hospital-at-Home [HAD], Coordinated Care Support [DAC]).

### Month-3 DH visit

At month 3, patients underwent an in-person DH reassessment following the same protocol as the initial visit. This included a complete physical examination.

## Data collection

At inclusion, patients completed a questionnaire (Annex 1) comprising different types of items: single-choice questions (demographic data) and five-point Likert-type scales. The Likert scale is designed to assess the degree of agreement with a given statement (e.g., from "strongly disagree" to "strongly agree"). These formats aim to better capture patients' opinions, perceptions, and preferences. They offer the advantage of being quick to complete, which is particularly relevant for patients who are often fatigued.

The survey focused on several key domains:

• General perceptions of telemedicine (feasibility)

• Perceived relevance and suitability of telemedicine for their individual care pathway

• Relationships with specialist physicians

• Relationships with relatives

These elements provide insight into the human and social dimensions of patients' perspectives on this mode of follow-up care.

Patients could skip questions or stop the questionnaire at any time. No identifying information was collected; each questionnaire was linked to a patient only by a study ID number.

During teleconsultations, physicians recorded key observations (e.g., connection quality, appointment delays, perceived care quality).

At the month-3 DH visit, a semi-structured interview (Annex 2) was conducted to assess patients' experiences with teleconsultations. Interviews were audio-recorded, transcribed verbatim, and analyzed qualitatively (sociological thematic analysis). Eleven patient interviews were included.

## Sample description

Each participant received a unique identification number linked to their completed paper questionnaire. This identifier allowed tracking of teleconsultation completion and, when applicable, association with the month-3 interview transcript. Only the principal investigator (VMS) and research coordinator (BC) had access to the linking file between patient identities and study data.

The questionnaire data analyses focused on patients' initial apprehensions and understanding of teleconsultations. Responses were described according to patient characteristics (age, place of residence, educational level, socio-professional category, etc.). For the subset of 11 patients with interviews, responses before teleconsultations were compared to those from the month-3 semi-structured interview. Statistical analysis consisted of simple descriptive statistics (frequencies, means) and cross-tabulations.

Raw questionnaire data were also coded using IRaMuTeQ software (IRaMuTeQ R 3.1.2, free version of ALCESTE) [22]. For each participant, gender, age at inclusion, employment status, education level, place of residence, and presence of a caregiver were recorded. Selected questionnaire responses (e.g., apprehension or interest in teleconsultations [yes/no], need for caregiver assistance to connect [yes/no]) were also included, along with the number of teleconsultations completed and interview corpus when available. IRaMuTeQ provided distributional statistical analyses and significant word occurrence frequencies within the corpus.

Item-by-item correlation analyses were performed with Jamovi software [23], which processes IRaMuTeQ data. Analyses were verified by the study epidemiologist (SL). Correlations between favorable (items 2, 3, 4, 5, 11, 12, 13, 14, 18,) and unfavorable (items 1, 6, 7, 8, 9, 10, 13, 15, 16, 19) attitudes toward teleconsultations were examined.

 

## Qualitative analysis

A sociological thematic analysis [24] and an IRaMuTeQ analysis were conducted on the 11 interview transcripts. IRaMuTeQ did not identify significant text clusters. The thematic sociological analysis was performed manually by transposing the corpus into a set of representative themes in relation to the research question. The analysis was conducted collaboratively by the research team, including a sociologist (AFR), an ethicist (BC), the head of the palliative care unit (NCH), and physicians (VMS, MB). The thematic analysis combined quantitative IRaMuTeQ/Jamovi findings with interpretative thematic analysis. No additional software was used.

## Ethics statement

This study was approved by the Ethics Committee of the Federal University of Toulouse Research Committee on February 17, 2022 (approval number: 2022−473). Written informed consent was obtained from all participants prior to study inclusion. Participants were informed that they could withdraw from the study at any time without any impact on their standard medical care.

## Results

### Demographics

Study participants are summarized in Tables 1 and 2. The study was offered to 38 successive patients; 2 were not included because they did not have access to a connected device. Twenty-nine women and seven men were included. The median age was 63.5 years (interquartile range: 52–72). Eighty-six percent of participants had a caregiver (either living as a couple or with a close relative). Fifty-three percent lived more than 50 km from the care facility. The 36 patients presented primary cancers of various origins, with a high proportion of breast (n = 11) and gynecological cancers (n = 6). Other cancers included squamous cell carcinoma (n = 5), metastatic melanoma (n = 4), lung cancer (n = 3), and gastrointestinal stromal tumor (n = 2), among others. Eighty percent had a good education level (equivalent to a French baccalaureate (High School diploma or higher), and 72% reported a comfortable living standard. All participants used a smartphone,

**Table 1. Demographic characteristics of the study population.**

| A: | | | | | |
|---|---|---|---|---|---|
| Gender | Women | Men | | | |
| | 29 | 7 | | | |
| **B:** | | | | | |
| Age at inclusion | < 30 years | 30-60 years | 60-80 years | > 80 years | |
| | 1 | 15 | 16 | 4 | |
| **C:** | | | | | |
| Living conditions | Living alone | Living with a partner | Living with a close family member | | |
| | 5 | 25 | 6 | | |
| **D:** | | | | | |
| Distance from care facility | <10 km | 10 - 50 km | 50 - 100 km | <100km | |
| | 7 | 10 | 12 | 7 | |
| **E:** | | | | | |
| Professional status | farmer | Craftman/trader, business Owners | Managers and higher intellectual professions | Intermediate occupation | Employee, workers |
| | 1 | 2 | 11 | 16 | 6 |

**Table 2. Baseline characteristics of patients.**

| Gender | Age | TC | DH Month 3 | Previous clinical trial | Highest degree obtained | Professional status | Previous telemedicine experience | Living conditions | Distance from care facility | Primary pathology |
|--------|-----|----|-----------|------------------------|-------------------------|---------------------|----------------------------------|-------------------|----------------------------|-------------------|
| male | 59 | 1 | no | yes | HSD | inter | no | partner | 43 | melanoma |
| female | 52 | 0 | no | yes | HSD | inter | yes | partner | 85 | breast cancer |
| female | 52 | 1 | no | no | bachelor | manager | yes | partner | 45 | rhabdomyosarcome |
| female | 65 | 2 | no | yes | HSD | inter | no | partner | 90 | gastrointestinal stromal tumor |
| female | 69 | 1 | no | yes | HSD | inter | no | partner | 140 | lung cancer |
| female | 69 | 0 | no | yes | YT NVQ | inter | no | partner | 20 | ovarian cancer |
| man | 69 | 2 | yes | yes | bachelor | manager | no | partner | 7 | mesothelium |
| female | 51 | 2 | no | yes | HSD | inter | no | relative | 45 | squamous cell carcinoma |
| female | 82 | 2 | Yes* | no | YT NVQ | employee | no | relative | 100 | melanoma |
| man | 76 | 0 | no | yes | PhD | manager | yes | partner | 10 | lung cancer |
| female | 47 | 1 | no | no | bachelor | manager | no | partner | 20 | breast cancer |
| man | 60 | 2 | yes | no | HSD | inter | no | partner | 35 | melanoma |
| female | 77 | 1 | no | yes | none | farmer | no | partner | 60 | squamous cell carcinoma |
| female | 62 | 0 | no | no | none | employee | no | relative | 100 | squamous cell carcinoma |
| female | 68 | 2 | Yes* | yes | HSD | manager | no | partner | 100 | breast cancer |
| female | 45 | 2 | no | no | bachelor | manager | no | partner | 70 | breast cancer |
| female | 72 | 2 | Yes* | no | HSD | inter | no | partner | 60 | rhabdomyosarcome |
| female | 37 | 2 | Yes* | yes | bachelor | inter | no | partner | 30 | breast cancer |
| female | 52 | 1 | no | yes | HSD | craftsman | no | partner | 50 | squamous cell carcinoma |
| man | 26 | 0 | no | yes | HSD | inter | yes | relative | 160 | medulloblastoma |
| female | 65 | 2 | Yes* | yes | bachelor | craftsman | no | relative | 80 | cervical cancer |
| man | 75 | 2 | no | no | bachelor | manager | no | partner | 30 | squamous cell carcinoma |
| female | 54 | 2 | Yes* | no | HSD | manager | yes | partner | 11 | breast cancer |
| man | 71 | 1 | no | no | none | worker | no | partner | 80 | gastrointestinal stromal tumor |
| female | 76 | 2 | no | yes | bachelor | inter | no | partner | 54 | ovarian cancer |
| female | 46 | 0 | no | no | bachelor | manager | no | aone | 100 | breast cancer |
| female | 68 | 1 | no | no | HSD | employee | no | partner | 86 | melanoma |
| female | 82 | 2 | Yes* | no | PhD | manager | no | alone | 8 | breast cancer |
| female | 51 | 2 | Yes* | yes | HSD | inter | no | relative | 60 | ovarian cancer |
| female | 70 | 2 | no | yes | HSD | inter | no | alone | 80 | cervical cancer |
| female | 87 | 2 | no | no | HSD | inter | no | partner | 10 | ovarian cancer |
| female | 84 | 2 | Yes* | no | YT NVQ | employee | no | alone | nd | ovarian cancer |
| female | 55 | 0 | no | no | HSD | inter | yes | partner | 20 | lung cancer |
| female | 53 | 2 | Yes* | no | YT NVQ | employee | yes | partner | 10 | breast cancer |
| female | 51 | 0 | no | no | bachelor | inter | no | partner | 100 | breast cancer |
| female | 68 | 2 | Yes* | yes | HSD | manager | no | alone | 145 | breast cancer |

**SD**: High School Diploma; **YT NVQ**: Youth training NVQ (National Vocational Qualification) level 1 et 2; **TC**: Number of teleconsultations performed; **DH Month 3**: Day Hospital (DH) Month 3: patients who benefited from a day consultation at 3 months. * indicate patients who agreed to the interview at 3 months.

tablet, or computer at least once per day, and 20% had prior teleconsultation experience. All patients with access to a connected device who were offered the study agreed to participate.

## Acceptability of telemedicine prior to the first consultation

Following inclusion, on the day of their first consultation with the Palliative Care Mobile Team (PCMT), patients completed a questionnaire assessing expectations and concerns regarding teleconsultations (Annex 1). The aim was to evaluate patients' general and personal attitudes toward telemedicine. In cases of fatigue, patients were instructed to respond only to the items they considered most relevant.

**A. Acceptability and understanding of the system.** All patients offered teleconsultation understood the process and were favorable to it. Indeed 100% of patients offered monthly telemedicine follow-up instead of quarterly in-person visits, and with the means to connect, accepted. The most frequently reported motivation was (i) the opportunity to consult a physician without the need to travel. (Ninety-four percent (32/34) agreed with the statement, *"I think telemedicine avoids the need to come onsite for medical questions"* (Fig 1A), ii) The reduction in fatigue associated with not having to travel. (Eighty-five percent believed remote appointments would reduce fatigue (Fig 1B)), and iii) The desire not to be a burden on one's relatives. (82% felt it would reduce inconvenience for their entourage (Fig 1C)). Only three patients responded to the item about reassurance against viral contamination or fatigue, limiting assessment of concerns regarding hospital visits (e.g., COVID).

All patients cited expectations of better follow-up (monthly instead of quarterly) and reassurance from more frequent physician contact as reasons for acceptance. Preferences for monthly in-person consultations were not explored.

**B. Perceived limitations of teleconsultations as reported by patients:.** *B.1. Perceived benefit for their follow-up care*: Ninety-four percent of patients considered telemedicine potentially useful, but only for certain medical activities (Fig 2A). Fifty-seven percent (16/28) disagreed that *"telemedicine is not suitable for my medical follow-up"* (Fig 2B). Only 50% believed teleconsultations would be as beneficial as in-person visits (Fig 2C).

*B.2. Potential concern regarding the therapeutic relationship:* Three items assessed whether patients feared dehumanization of medicine (absence of tactile or olfactory contact and the fact that patients had not previously met the palliative care physicians.). We included three items on this topic on which patients were invited to express their views.

1. Telemedicine dehumanizes medicine

2. Telemedicine weakens relationships with caregivers

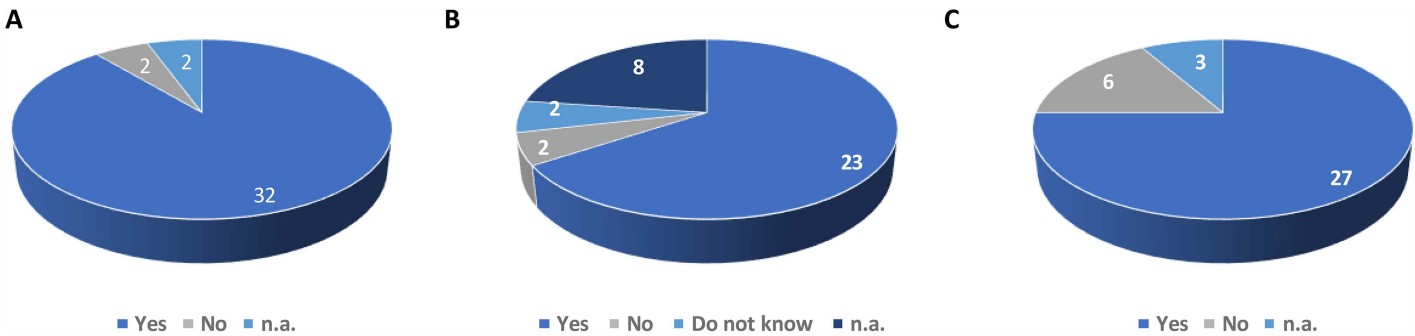

**Fig 1. Perceived benefits of teleconsultations for newly enrolled palliative care patients.** A: Patients' responses to the item: "I think telemedicine prevents me from having to come on site to address my medical concerns." B: Patients' responses to the item: "Telemedicine will save me time." C: Patients' responses to the item: "Telemedicine makes it easier for my relatives to accompany me to consultations compared to day-hospital visits." n.a.: patients did not answer the question.

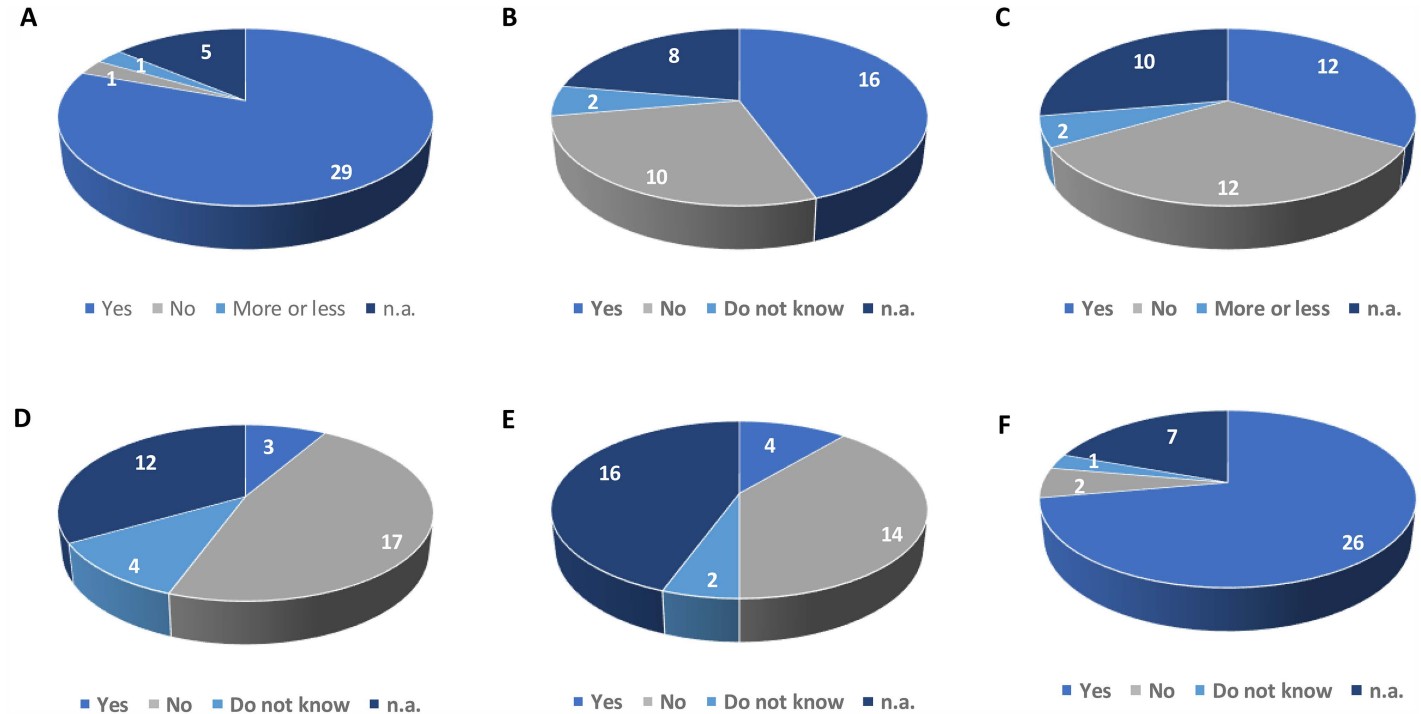

**Fig 2. Perceived limitations of teleconsultations for newly enrolled palliative care patients.** A: Patients' responses to the item: "I think telemedicine could be useful, but only for certain medical procedures." B: Patients' responses to the item: "I think telemedicine is not suitable for my medical follow-up." C: Patients' responses to the item: "A teleconsultation is as beneficial as an on-site consultation." D: Patients' responses to the item: "Telemedicine dehumanizes medical care." E: Patients' responses to the item: "Telemedicine weakens relationships with healthcare providers." F: Patients' responses to the item: "Monthly telemedicine allows maintaining a strong connection with the care team." n.a.: patients did not answer the question.

3. Monthly telemedicine maintains a strong link with the care team

Only 12% (4/24) agreed that telemedicine dehumanizes medicine (Fig 2D), 70% (14/20) felt it did not weaken relationships (Fig 2E), and 93% (29 respondents) believed it maintained a strong link (Fig 2F).

**B.3. Concerns regarding their ability to connect**: We investigated the anticipated practical aspects of teleconsultations. The included patients were of varying ages, and some had limited experience with digital technologies. We first assessed potential concerns patients might have regarding the conduct of teleconsultations. For example, we asked whether they were worried about their ability to connect (i.e., their proficiency with digital tools) and whether they were concerned about not being able to hear the physicians clearly.

For the item, *"I am afraid I will not understand how to connect,"* 93% of respondents (25/27) expressed confidence in their ability to connect, as they responded negatively to this statement. However, approximately one-third of respondents (30%) indicated that they would prefer assistance, as 8 out of 27 answered positively to the item, *"I would prefer to have someone help me connect."*

All respondents had no concern about hearing or understanding the physician (Fig 3C). No patient reported fears regarding data security or unauthorized recording.

**C. Correlation analyses.** No correlation was observed between gender, residence, education, or profession and attitudes toward teleconsultation (p > 0.7). A slight correlation existed between age and apprehension, with older patients expressing minor concerns about needing help (P = 0.061).

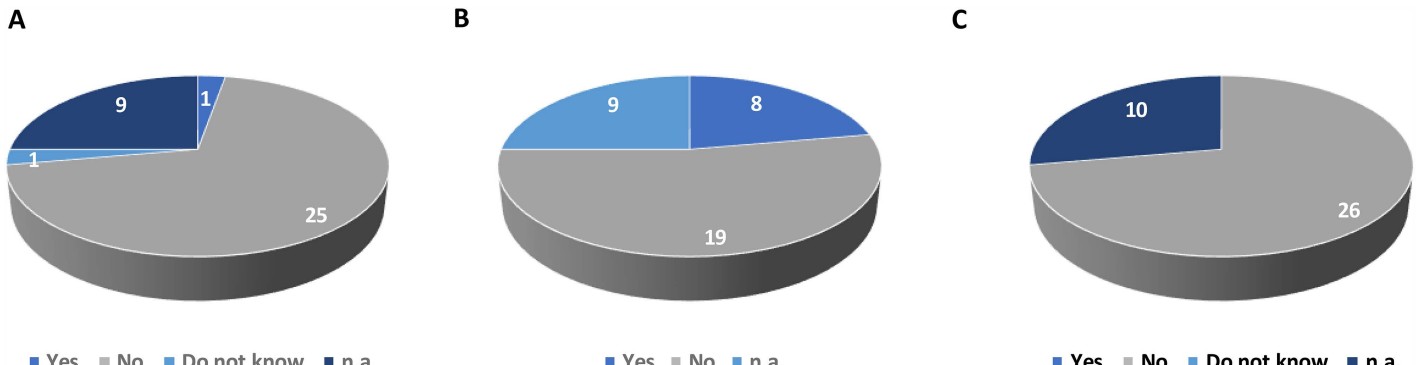

**Fig 3. Potential patient concerns regarding their ability to use digital tools.** A: Patients' responses to the item: "I am worried I won't understand how to connect." B: Patients' responses to the item: "I would prefer to have someone help me connect." C: Patients' responses to the item: "I am worried I will not hear (or understand) what the doctor says."n.a.: patients did not answer the question.

Overall, the offer of monthly teleconsultations was well received. Patients were confident in this follow-up modality, though they acknowledged it could not address all medical issues. Teleconsultations were presented as an adjunct rather than a replacement.

### C.1.  Feasibility of teleconsultation – conduct of the consultations

Seventy-two teleconsultations were initially planned (two per patient before the 3-month day hospital visit). A total of 48 teleconsultations were completed; Despite the inclusion criterion of life expectancy ≥3 months, only 13 of 36 patients included in the study completed the study.. Others died before the first teleconsultation (n=8), between the first and second (n=8), or before the day hospital visit (n=7) (Fig 4A, Table 2). We were therefore only able to schedule 48 tele-consultations, as 64% of the enrolled patients died before completing the three-month study period (This highlights late referral to palliative care [25]. Fif0074y percent had previously participated in at least one clinical trial; no correlation was found between trial participation and 3-month survival (p>0.5). This finding highlights that patients are referred far too late to palliative care teams in their care trajectory [27]. Fifty percent of them had participated in at least one clinical trial during their disease course. We did not observe any correlation between participation in a clinical trial and 3-month survival (p>0.5).

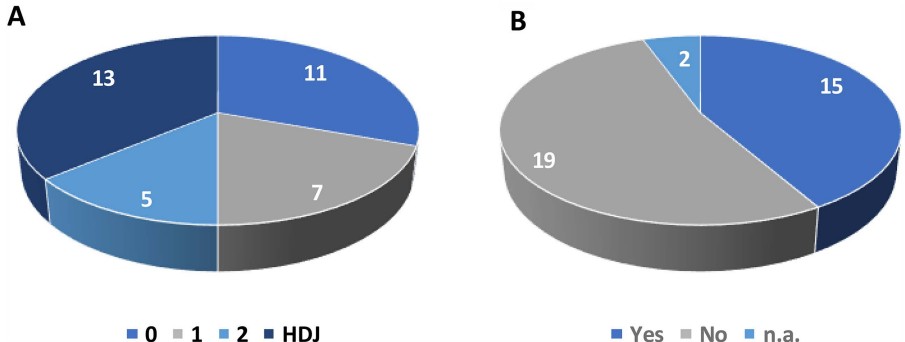

**Fig 4.  A: Distribution of patients according to whether they completed 0, 1, 2, or 3 consultations (HDJ). B: Number of patients who participated in a clinical trial before being referred to the Mobile Palliative Care Team (PCMT). n.a.: patients did not answer the question.**

Of the 48 scheduled teleconsultations, all 48 were completed within the framework of this project, including two that were postponed by a few days. Overall, patients experienced no significant difficulties connecting (only two connection postponements throughout the study, one of which was due to the patient being on vacation in a location without internet access). Most patients used a computer; no delays were observed. No gender or age differences were noted; even patients over 80 connected successfully. Most connected with a caregiver. Education level and social status did not affect digital tool usage. Teleconsultations are therefore feasible for all patients. Consultations lasted 20–35 minutes. They allowed global health assessment and prescription of medications, delivered by email to patients or pharmacies.

Some teleconsultations made it possible to improve uncomfortable symptoms: for two patients who presented an increase in pain, the management consisted in an adaptation of the analgesic treatment for one and in the introduction of a step III analgesic for the other. The advantage of these two teleconsultations compared to a simple telephone call was that they made it possible to see how the patient was moving and gave indications regarding the etiological diagnosis (despite the absence of a physical examination). A prescription was then sent to the pharmacy and to the patient.

For another patient, in view of the deterioration of her general condition and the difficulties of staying at home, the implementation of a home hospitalization service was proposed: the patient was able to die at home, without transfer to the emergency department or to an inpatient unit (which was not desired by the patient). For another patient, the "systematic" teleconsultation made it possible to suggest the diagnosis of a femoral fracture: she was able to have simple radiographic images taken, confirming the fracture, then her case was directly discussed in the pain management and bone metastases multidisciplinary team meeting (MDTM), and she was directly admitted to orthopedic surgery, without going through the emergency department, which would not have been appropriate in this palliative context. Finally, for another patient, the visualization of the tumor wound with a lot of exudate made it possible, in collaboration with the community nurse present at home during the teleconsultation, to adapt the protocol, and thus the comfort of the patient.

Compared to these situations, in the absence of appropriate treatments or modifications to management, these patients would ultimately have been hospitalized, or even ended up in the emergency department, which is not suitable in the palliative situation and not necessarily desired.

Moreover, in oncology, due to this long medical pathway, very often, patients "wait" until the situation is extremely deteriorated before finally turning to the emergency department, even if this is not appropriate. It can be noted that for the above-mentioned patients in our study, the systematization of teleconsultations made it possible to bring to light symptoms even though the patients had not contacted the mobile palliative care team, although they had the contact details. Each issue was able to be addressed, and this made it possible to improve their quality of life. However, we are aware that, due to our limited sample size, these elements are only avenues for further research.

### C.2. Evaluation of teleconsultations

Eleven of 13 patients evaluated at 3 months agreed to interviews (Annex 2). Two were too fatigued to participate. All interviewed patients were satisfied with teleconsultations, highlighting reassurance from monthly physician contact. Three revisited concerns about potential dehumanization, but all considered teleconsultations valuable. Regarding frequency, 64% preferred two teleconsultations per in-person visit, while 36% preferred alternating teleconsultation and day hospital visits. No patient requested more teleconsultations.

### C.3. Analysis of free-text patient responses

**1. Connectivity and mobility: material and relational resources.** The patients interviewed emphasized the benefits of teleconsultation in terms of time savings and, above all, improvements in quality of life (less fatigue). Indeed, with regard to the physical consequences of traveling—which can lead to an increase in pain—we confirm that patients (or their caregivers) express that, in palliative care, access to medical services can be challenging due to the physical strain caused by illness:

*"Sometimes you have to go see a doctor, but I think it really helps for people like my mother, who are very tired, to avoid making trips (…) just to check that everything is fine. It saves having to travel back and forth to Toulouse and tiring the person out; teleconsultations like this are great, in my opinion."*

*"Not having to travel is important (…) I have pain in my legs (…) I appreciate the fact that there is less travel involved."*

Overall, patients overwhelmingly endorsed the fact that monthly teleconsultations made it possible to maintain medical follow-up without the fatigue or pain associated with travel. We did not observe any correlation between satisfaction with avoiding travel and the distance between the patient's home and the healthcare facility.

One of the objectives of the project was to determine whether there is a particular patient profile that could benefit from teleconsultations, with questions regarding age, social status, and ease with handling digital tools. Indeed, since teleconsultations require suitable computer equipment, they contribute to forms of inequality in access to healthcare by reproducing, in this context, existing digital inequalities. In our study, 2 patients could not be included in the protocol because they did not have any digital devices.

However, most people said they connected easily by themselves (*"it's very simple, I think I even received the text message on the phone plus the email to my email address; you just had to click on the link, and it sent us directly to the page"*). Qualitative analyses, however, show that the mediation of a third party often plays a role. Indeed, for patients with limited computer skills, relying on a relative to facilitate the connection can be central to carrying out teleconsultations. As shown by this interview excerpt, access to teleconsultation should be understood in relational terms (the patient's relational resources) and in temporal terms (the timing of the connection):

**Patient** *"It's going well. You need someone who knows how to handle the computer."*

**Researcher** *"But your husband is there. And you need to be equipped."*

**Patient** *"On my own, I wouldn't have done it."*

**Researcher** *"The nurse might have helped you."*

**Patient** *"Maybe. But given the time of the appointment, the nurse would have to be there."*

For this patient, her spouse constitutes a major resource allowing her to carry out the teleconsultations.

Another patient calls on her daughter to connect to the platform. When asked whether it was easy for her to connect, she replied: *"Well, you'd have to ask my daughter that."*

**2. Teleconsultation and dehumanization of medicine.** Although the responses to the inclusion questionnaire did not reveal any concern about a dehumanization of medicine, we asked about this item during the final interview.

The patients met the palliative care doctor once at the day hospital before the proposal of teleconsultation. In this configuration, the phase of getting to know each other and its conditions are reduced regarding the establishment of a relationship of trust. Nevertheless, all the patients accepted the following consultations by videoconference. According to what the interviewed people said, these first in-person meetings, even if limited, allow an easier transition to teleconsultation:

*"The fact that I had met her once, so I had a sure face (…) I felt confident right away."*

*"Because it's a question of trust (…) of having already talked with [her]."*

We asked about the "practical" side of teleconsultations (access to medications for example). The patients did not mention any difficulty. We can cite a patient speaking about the circulation of prescriptions:*"Yes, she also sent them directly to the pharmacy, there were some prescriptions that needed to be encrypted."*

Still linked to our concern of determining whether there was a patient profile suited to teleconsultations, we asked about the patients' ease in talking with their doctor in teleconsultation. Indeed, the stakes of interactions between doctor and patient must be examined in light of their social background, because the abilities to put one's own body and bodily sensations into words are socially constructed and situated, and vary with the level of health literacy. People with "low" literacy have more difficulty describing their symptoms, expressing their concerns and carrying out self-management of the illness [26,27].

The people we interviewed had school and cultural capital, strongly connected to health literacy, built up in long patient careers. We can thus note a certain fluidity in the exchanges during teleconsultations:

**Researcher** *"Were you embarrassed to ask questions?"*

**Patient** *"No. I tell you: the bonds have strengthened!"*

**Researcher** *"Is that because of the doctor's personality or the screen?"*

**Patient** *"A bit of both, I think. I would say both, because I am easy to approach, Ms. X too, and well, she hears everything you tell her."*

If for some bodily signs the demonstration or exposure is easier (*"Yes, I was also able to show her my tumor wound"*), for other signs more complex to show or imitate, the exercise requires a particular performativity from the patients. Indeed, expressing one's symptoms or questions requires an ability to put into words the body and its sensations. A woman comes back to the differences between in-person and video consultations, in particular with regard to the difficulties of verbally expressing bodily signs that are invisible for the camera:

*"It wouldn't have been easy. Because it's not very big and it's skin-colored. (…) So there are differences. Teleconsultation, I would say, is good to make up for it a bit but it's not worth the real consultation."*

Some interviewed patients also express the need for a longer time for the teleconsultation to obtain information and/or to put into words their various bodily feelings:

*"I found that almost 20 minutes was too short (…) maybe even to do it longer."*

While the patients say that demonstration is possible (*"part of my body [that] needed to be seen!"*), the visualization of the body asks of them a performativity and an attitude that must be even more active in teleconsultation:

*"So I stood up, we stood up together and I said 'there it is (the pain)'."*

It is once again important to situate these patients socially, possessing school and medical capital, and who say they are comfortable explaining their feelings to doctors via a screen. To the question of whether they felt any difficulties explaining their health problems, the majority answered negatively, as expressed by this patient:

*"No, not at all, since we have the face and we express what we want."*

## Discussion

An increasing number of patients with advanced cancer but a life expectancy greater than six months will be referred to palliative care mobile teams (PCMT) in the coming years. Advances in anticancer treatments have allowed many patients

to transition to a chronic disease trajectory, where they are followed for several months or even years without curative intent. French hospitals are considering follow-up strategies for these patients, initially in parallel with the referring oncologists and palliative care teams, and later exclusively by palliative care teams. The current trend is to integrate palliative care physicians early in the patient pathway, particularly in gynecologic oncology, to prevent discontinuities in care when specific curative treatments are stopped. Deprescription is also increasingly emphasized to improve end-of-life quality of life and reduce aggressive interventions in the last two months of life (ASCO recommendations). This trend will likely increase the workload for palliative care teams in the medium term.

Currently, due to limited human resources and overloaded facilities, our center can offer only quarterly follow-up for patients exclusively managed by palliative care physicians. However, patients in palliative care often experience rapid deterioration between in-person visits and may require urgent hospitalization or intensive care, where maximalist interventions are systematically proposed. Such care is inconsistent with policies prioritizing quality of life at the end of life and discouraging aggressive interventions during the final month.

In this context of high demand for palliative care and limited bed availability, teleconsultations appear as a potential support for cancer patient follow-up. They are not intended to replace in-person consultations but rather to provide supplementary support and maintain a strong connection between the hospital care team and the patient.

The objective of this qualitative study was to assess whether teleconsultation follow-up could be offered to all patients regardless of gender, age, digital literacy, social status, place of residence, or prior perceptions of telemedicine. Patients recruited for this study met the PCMT for the first time when teleconsultations were proposed. We sought to determine whether trust necessary for telemedicine follow-up could be established in a single in-person consultation. Newly referred patients often come from a curative project or clinical trial and are vulnerable, anxious about disease progression, yet experienced regarding their prolonged care pathway.

## Recruitment and sample characteristics

This study included a small sample of 36 patients. Therefore, the quantitative descriptive analyses should only be considered as illustrative of the results of the qualitative analyses, which were the core of this work. The sample includes a majority of women (75%). We cannot explain this gender difference by life expectancy or recruitment bias, as all eligible patients (except two) agreed to participate. The study was conducted in a center with high recruitment of breast and gynecological cancer patients, where oncologists are highly attuned to palliative care. This may explain the higher female representation and the chronicity of these cancers, allowing metastatic female patients to live longer. The fact that all interviews were conducted with women is a limitation, highlighting the need to consider gender differences in therapeutic relationships—both in-person and remote—and how they intersect with social class. Relationships to pain and the body differ by gender [28,29], as do interactions with the medical world, especially in oncology [30]. Gender differences are also evident in supportive care participation, including teleconsultations, where the dynamics of "saying," "showing," and "doing" differ.

Dudoit et al. (2007) [31] showed that women tend to actively "contribute" themselves and their bodies in supportive oncology care, facilitating touch and discussion of personal experiences. In men, the dominant model is more about "making their body available," closer to a body-object than a lived body, with less mediation between body and discourse. These mediation forms are crucial when consultations occur via a screen, and our results must be considered in the context of a female-only sample. Future studies should include a higher proportion of male patients.

## Social status and digital access

Most patients had a comfortable social status. In France, all citizens have free access to healthcare, including palliative care, so this bias is not explained by income. Gender and social status may influence longevity in oncology care; women with higher social status may better endure the demands of treatment plans. Patients with lower incomes tend to have shorter life expectancy, potentially limiting access to palliative care.

Teleconsultations aimed to reduce healthcare access inequalities. Two out of 38 patients lacked the means to connect, indicating that approximately 5% of patients would be unable to benefit from teleconsultations. Offering teleconsultations in local pharmacies equipped with telemedicine booths could overcome this limitation, providing access for patients without personal digital devices and offering connection assistance.

## Technology and relational resources

From the equipment perspective, even if all patients say they have a computer or a phone, as well as a stable internet connection, access to technology is not determined by this factor alone. Indeed, a distinction appears between the possession of the necessary computer equipment and the practical knowledge of how to use it [32].

*"Last time, what I didn't understand is that I did it with the iPhone (…) and when I had the consultation with Doctor X it was impossible to have the camera, I couldn't see her or I could see her but I couldn't hear her, there was something that didn't work, I didn't understand why, especially since it worked before."*

Thus, when setting up teleconsultations, beyond the material and practical conditions linked to digital tools, our study shows that the patients' social or relational capital must be taken into account. Nevertheless, the relational dimension, which can be mobilized as a resource allowing one to call on a mediator with the required technological skills, can also become a constraint. Indeed, in this case, access to the consultation depends on the availability of these third parties. For example, for one patient, teleconsultation appointments had to be rescheduled to a time slot when her daughter was available (outside her working hours). The role of caregivers is therefore central in access to care and in evaluating the uses of telemedicine.

Finally, the need for the presence of a third party raises the question of the private nature of exchanges between doctor and patient. First, the presence of caregivers for setting up the teleconsultation leads us to wonder whether they remain in the same space in order to provide support to the patients. Then, while most of the patients interviewed can isolate themselves in their home (*"my husband was there, but I was in my office"*) for the teleconsultation, we can assume that this is not necessarily the case for patients in certain family and social configurations (crowded households, single-parent families, young children, etc.). These conditions of realization can affect the transmission by patients of certain important information for supportive care follow-up, on the one hand through the distraction that third parties can bring, and on the other hand through a restricted freedom of speech. Here too, we can therefore mention the relevance for patients of using teleconsultation booths, available free of charge in pharmacies.

## Patients' Perceptions and Adaptation

At inclusion, no correlation was observed between gender, residence, education, or profession and initial positive opinion on teleconsultations. Older patients showed slight concern about digital skills, but most patients (93%) were confident. Data security and understanding of the physician were not reported as concerns. However, the study population had an average or higher social status, representing a limitation.

Patients initially struggled to imagine teleconsultation potential, with mixed or even contradictory responses. About one-third felt telemedicine dehumanized medicine (Fig 2D), yet 93% believed it maintained a strong connection with the care team (Fig 2F). Subsequent adherence was high.

Patients' adaptation to teleconsultations, despite limited prior interaction with healthcare providers, can be understood in the context of their position within the care pathway. Indeed, engagement in palliative care presupposes a prior history of the disease and its management. In the interviews, teleconsultations occurred at a stage when patients had already established a network of relationships with medical staff. Palliative care typically involves individuals experiencing disease recurrence and/or considered to be in a state of oncological chronicity. These patients often possess accumulated

experiential knowledge about their illness and about the technical and professional environment of their care (so-called expert patients). Furthermore, some had previously participated in other clinical trials (Fig 4B).

These elements situate patients within what Anselm Strauss (1992) [33] terms a "disease trajectory," which assumes a professional dimension or "sick role career." This encompasses both the physiological progression of the disease and the entire organization of work involved in monitoring it, by all actors, including the patient. For patients and their families, this trajectory constitutes a social experience of the bodily manifestations of illness (discomfort, fatigue, pain) as well as interactions with healthcare professionals and coordination work among multiple actors. Teleconsultations thus represent an additional resource within their trajectory and their own "patient work." Some patients even proposed using teleconsultations for non-therapeutic objectives, such as care coordination:

*"Ah yes, yes, it's part of the same process, maybe in a somewhat more collegial way, we could meet together, several people; well, maybe that could be interesting (…) I think it could even be a bit of a collegial meeting, but without being too pushy either."*

Some interviews occurred in the presence of a healthcare professional, for example, a home nurse during a dressing change, which allowed modification of the dressing protocol.

For patients familiar with telecommunication tools, such interactions can streamline the care pathway, particularly in the management of prescriptions:

*"Yes, she also sent them directly to the pharmacy; some prescriptions needed to be encrypted."*

For certain individuals, these experiences in palliative telemedicine contribute to what could be described as a "tele-sick role career," running parallel to in-person consultations while forming part of the overall patient trajectory. Most interviewed patients had previous experience with teleconsultations. In this way, the use of digital tools in care constitutes a new form of home-based patient work [34].

*"I like it because now I have difficulty moving around due to the illness, so it allows me to alternate; every other consultation I do via teleconsultation with my primary care physician."*

Qualitative analysis of patients' open-ended reflections on teleconsultations shows that they are capable of organizing their own care and developing self-monitoring practices. This raises the question of whether the social characteristics of the patients surveyed allow them to be framed as "sentinel patients." Patrice Pinell (1992) [35] introduced this term in oncology to describe individuals who can clinically observe their own body and alert the physician when an early or recurring symptom appears. Telemedicine thus presupposes that patients have the capacity to provide information to the physician—describing symptoms and physiological states—a socially situated skill, influenced by factors such as gender and social class.

*"Well, when I described my pain (…) the doctor said, 'If you are suffering like this, this is not normal pain; I think there is still something in the bone causing your pain' (…) And indeed, the bone was broken again. During the consultation she asked, 'Are you in pain right now?' So I stood up, we stood up together, and I said, 'Here it is.' And indeed, that was it."*

An additional observation is that teleconsultations are often perceived less as "formal consultations" and more as a channel for maintaining the patient–physician relationship:

*"I like it (…) it allows me to alternate; every other consultation I do via teleconsultation with my primary care doctor."*

Thus, teleconsultations are seen by patients as a relational resource with healthcare professionals. The fluidity of telemedicine interactions should be considered in the context of existing smooth in-person therapeutic relationships, but also in their social context. This "singular colloquy" between patient and physician, mediated through speech [36], favors individuals with medical cultural capital—those able to verbalize their condition without challenging medical authority. Teleconsultation ultimately mirrors the sociological dynamics of in-person consultations. Social science research shows that care is often most effective with patients who adhere to medical norms, seek to understand their care, ask sufficient questions for treatment follow-up, yet do not challenge medical authority. These are the patients who appear to constitute the study population of Telemsos. For them, teleconsultations represent opportunities within the care pathway. They are embedded in "sick role careers," possess substantial medical and cultural capital, including high health literacy, and actively position themselves as "sentinels" of their own health.

### Infection risk, quality of life, and caregiver burden

Regarding infection risk, most patients did not prioritize telemedicine for enhanced safety, possibly due to confidence in oncology services. Teleconsultations may reduce patient fatigue (85%) and caregiver disturbance (82%), helping prevent caregiver burnout. Abbes et al. (2021) reported that among 121 caregivers of cancer patients, 67.7% experienced moderate-to-severe burden, with over half exhibiting anxiety (59.5%) or depression (57%) [37].

Finally, regarding quality of life and the patient's social environment, teleconsultation appears to have an impact both on the patient and on caregivers. For 85% of patients, it helps reduce fatigue by limiting travel, and for 82% of caregivers, it reduces the disruption to their daily lives. This factor should be considered to help prevent caregiver burnout. Indeed, Wafa Abbes [37] studied 121 caregivers of cancer patients, assessing their levels of anxiety, depression, and perceived burden. The study found that 67.7% of caregivers reported a moderate to severe sense of burden, according to the Zarit scale, and that more than half of the caregivers exhibited symptoms of anxiety (59.5%) and/or depression (57%), based on the HAD scale.

### Conclusion

This study evaluated whether age, gender, socio-economic status, or pathology influenced adherence to and feasibility of teleconsultation in palliative care. Within the limits of our sample, these factors did not affect the smooth conduct of teleconsultations. Patients were generally able to connect without major difficulty, with some receiving assistance when needed.

Teleconsultations provided clinical and psychological benefits and supported monthly follow-up. All patients reported positive outcomes, and a trusting relationship with their physician was established even without prior teleconsultation experience. Teleconsultations identified symptoms (pain, vomiting) and clinical changes (tumor progression, decline in general condition), enabling timely interventions that improved symptoms and quality of life. Some consultations facilitated home-based care coordination, avoiding unwanted hospitalizations, while others enabled direct hospital admission, bypassing emergency services.

Patients expressed a desire to continue monthly teleconsultations after the study period. These findings highlight the value of teleconsultations for palliative oncology patients, allowing more frequent follow-up for a larger patient population.

However, patients were referred late to palliative care. In this study, 64% of patients offered teleconsultations died within three months of inclusion, reflecting delayed access. These results are consistent with prior studies showing early palliative care improves quality of life, reduces depressive symptoms, and can increase survival [38–40].

Sociologically, three key points emerged:

1. Access to care should be considered in terms of connectivity, not just geography, incorporating technical, temporal, and relational factors.

2. Teleconsultation functions as a complementary resource within patients' broader care trajectories. The ability to utilize this resource is socially situated and depends on educational, cultural, and medical capital.

3. Patient capacity to communicate symptoms and engage in teleconsultations is influenced by social and gendered factors, highlighting the importance of social and medical literacy.

Following this study, monthly teleconsultation access has been institutionalized for PCMT patients. Multicenter studies are recommended to validate these findings and develop standardized palliative care models in oncology.

Early integration of palliative care remains critical. Beyond funding, this requires a shift in attitudes among healthcare providers and the public to recognize palliative care as more than end-of-life care. Structured models, such as Advance Care Planning, may facilitate this transition.

## Supporting information

**S1 File. Questionnaire: Questionnaire administered to patients on the day of their inclusion in the protocol.**
(DOCX)

**S2 File. Semi-Structured Interview Protocol: Outline of the semi-structured interview used by the investigators during the 3-month post-inclusion visit.**
(DOCX)

## Author contributions

**Conceptualization:** Bettina Couderc.

**Data curation:** Valerie Mauries-saffon, Marie Bourgouin, Bettina Couderc.

**Formal analysis:** Alfonsina Faya-Robles, Sebastien Lamy, Nathalie Caunes-Hilary.

**Funding acquisition:** Bettina Couderc.

**Investigation:** Valerie Mauries-saffon.

**Writing – original draft:** Valerie Mauries-saffon, Bettina Couderc.

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
