## [Decision Letter · Decision Letter 0]

23 Jan 2025

Dear Dr. Couderc,

Thank you for submitting your manuscript to PLOS ONE. After careful consideration, we feel that it has merit but does not fully meet PLOS ONE’s publication criteria as it currently stands. Therefore, we invite you to submit a revised version of the manuscript that addresses the points raised during the review process.

I agree with the reviewers. Please see the reviewers' suggestions and make the changes as suggested and answer the reviewers' questions.

We look forward to receiving your revised manuscript.

Kind regards,

Alexandre Morais Nunes, Ph.D.

Academic Editor

PLOS ONE

Journal requirements: When submitting your revision, we need you to address these additional requirements. 1. Please ensure that your manuscript meets PLOS ONE's style requirements, including those for file naming. The PLOS ONE style templates can be found at https://journals.plos.org/plosone/s/file?id=wjVg/PLOSOne_formatting_sample_main_body.pdf and https://journals.plos.org/plosone/s/file?id=ba62/PLOSOne_formatting_sample_title_authors_affiliations.pdf. 2. We note that your Data Availability Statement is currently as follows: [All relevant data are within the manuscript and its Supporting Information files.] Please confirm at this time whether or not your submission contains all raw data required to replicate the results of your study. Authors must share the “minimal data set” for their submission. PLOS defines the minimal data set to consist of the data required to replicate all study findings reported in the article, as well as related metadata and methods (https://journals.plos.org/plosone/s/data-availability#loc-minimal-data-set-definition). For example, authors should submit the following data: - The values behind the means, standard deviations and other measures reported;- The values used to build graphs;- The points extracted from images for analysis. Authors do not need to submit their entire data set if only a portion of the data was used in the reported study. If your submission does not contain these data, please either upload them as Supporting Information files or deposit them to a stable, public repository and provide us with the relevant URLs, DOIs, or accession numbers. For a list of recommended repositories, please see https://journals.plos.org/plosone/s/recommended-repositories. If there are ethical or legal restrictions on sharing a de-identified data set, please explain them in detail (e.g., data contain potentially sensitive information, data are owned by a third-party organization, etc.) and who has imposed them (e.g., an ethics committee). Please also provide contact information for a data access committee, ethics committee, or other institutional body to which data requests may be sent. If data are owned by a third party, please indicate how others may request data access. 3. Please include a copy of Table 1 which you refer to in your text on page 8.

Reviewers' comments:

Reviewer's Responses to Questions

**Comments to the Author**

1. Is the manuscript technically sound, and do the data support the conclusions?

Reviewer #1: Yes

Reviewer #2: No

2. Has the statistical analysis been performed appropriately and rigorously?

Reviewer #1: Yes

Reviewer #2: N/A

3. Have the authors made all data underlying the findings in their manuscript fully available?

Reviewer #1: Yes

Reviewer #2: No

4. Is the manuscript presented in an intelligible fashion and written in standard English?

Reviewer #1: Yes

Reviewer #2: Yes

Reviewer #1: The paper presents an interesting and relevant topic. However, it suffers from several drawbacks that need to be corrected before its possible acceptance.

- The abstract needs to be rewritten, especially clarify the background, after objectives.

- I advise the authors to professionally proofread their manuscript prior to resubmitting.

- Introduction is complet in the health area. However, I think it could better frame the problem under study, indicating more explicitly the factors that were at the basis of the research.

- Methods chapter needs more specification of empirical work. For example, I would like to see a better characterization of the target population and sample in the empirical part

- In structural terms, I suggest that the discussion should be combined with the results and that the conclusions should be in a “CONCLUSIONS” chapter.

Reviewer #2: The study identifies a predominance of affluent female patients, likely influenced by the institute’s focus on breast cancer. However, it does not adequately explore the implications of this demographic concentration. Provide detailed information on recruitment strategies. Specify whether efforts were made to include a broader demographic, such as patients from varied socioeconomic backgrounds, men, or those with other cancer types. Explicitly discuss how this sample demographic may limit the generalizability of findings, particularly for less affluent, male, or geographically isolated populations. If data permits, include subgroup comparisons (e.g., by gender, socioeconomic status, or geography) to identify variations in teleconsultation experiences or outcomes.

Exclusion of 5% of patients due to lack of internet access underlines inequities in teleconsultation availability. Suggest strategies such as providing technical equipment (e.g., tablets, mobile hotspots); establishing community telemedicine hubs for shared access; and offering telephonic consultations as a low-tech alternative. Discuss how these findings could inform policies aimed at reducing the digital divide, such as public-private partnerships to improve digital infrastructure in underserved areas or subsidies for patients to access necessary technology.

The study briefly mentions educational, cultural, and medical capital but lacks depth on their influence on teleconsultation outcomes. Incorporate specific patient stories or quotes from interviews that highlight how these forms of capital impact engagement and satisfaction with teleconsultations. Provide actionable insights on how teleconsultation practices can be adapted to different social profiles. For instance, patients with lower digital literacy may benefit from tutorials or simplified interfaces, while those with high medical literacy may prefer in-depth discussions.

The high mortality rate (64%) within three months of enrolment indicates delayed referrals to palliative care, limiting teleconsultation benefits. Investigate and discuss factors contributing to late referrals, such as healthcare provider hesitancy, lack of patient awareness, or systemic delays. Advocate for integrating palliative care earlier in the oncology care pathway, emphasising benefits beyond end-of-life care. Reflect on how late-stage enrolment may have constrained patients’ ability to adapt to teleconsultations, potentially skewing satisfaction or utility perceptions.

The absence of standardised quantitative outcome measures weakens the study's conclusions. Include quantitative tools such as patient satisfaction surveys for a standardised view of teleconsultation acceptance, health-related quality of life measures to evaluate broader impacts, and symptom management success rates or reduced hospitalisations. Discuss how these metrics corroborate or challenge qualitative findings, adding rigour to the study’s conclusions.

Gendered socialisation is mentioned but not explored in detail regarding its influence on teleconsultation outcomes. Elaborate on how gender roles and expectations may shape teleconsultation engagement. For instance, women may be more accustomed to caregiving roles, influencing their comfort with digital health tools, and communication styles may vary by gender, affecting the dynamics of virtual consultations. Provide evidence or examples from interviews or observations that highlight these gendered differences in teleconsultation experiences.

While identifying barriers like digital literacy and connectivity, the study does not propose concrete solutions. Discuss initiatives such as training programs for patients and carers to improve digital literacy and developing user-friendly telehealth platforms with intuitive interfaces. Highlight how healthcare systems and policymakers can address these barriers, including funding for digital tools or legislation to ensure equitable access to telemedicine.

The three-month study duration limits insights into the sustained impact of teleconsultations. Suggest follow-up studies spanning six months to a year to assess long-term outcomes, including patient satisfaction, clinical effectiveness, and cost-efficiency. Discuss potential issues in maintaining engagement over time, such as digital fatigue, evolving patient needs, or technological challenges.

**Do you want your identity to be public for this peer review?** For information about this choice, including consent withdrawal, please see our Privacy Policy

Reviewer #1: **Yes:** Andreia Matos

Reviewer #2: **Yes:** Ricardo de Moraes e Soares

---

## [Author Response · Author response to Decision Letter 1]

4 Mar 2025

Response to Reviewers and Revised Manuscript Submission

Dear Editor,

Please find attached our point-by-point response to the reviewers, along with the revised version of the manuscript, in which all the requested revisions have been addressed. We sincerely thank the reviewers for their insightful comments and suggestions, which have significantly improved the quality of our manuscript. We hope that the revised version now meets their expectations.

As the manuscript has been reviewed and corrected by a native English speaker, a substantial number of modifications have been made. To facilitate the reviewers' verification of the most critical changes, we are submitting both a version with tracked changes and a clean revised version.

We remain available for any further modifications if required.

Best regards,

Bettina Couderc

Reviewers' Comments and Responses

Reviewer #1: The paper presents an interesting and relevant topic. However, it suffers from several drawbacks that need to be corrected before its possible acceptance.

- The abstract needs to be rewritten, especially clarify the background, after objectives.

We have rewritten the abstract to better align it with the manuscript, as suggested by the reviewer.

Abstract:

Background: Advances in cancer treatments have increased life expectancy, leading to a growing need for palliative care services. Due to the saturation of hospital facilities, teleconsultation is emerging as an innovative solution to improve patient follow-up. This study aims to assess the acceptability and feasibility of monthly teleconsultations for chronically ill cancer patients newly referred to palliative care. Methods: Thirty-six patients were included based on predefined criteria, specifically ownership of a connected digital device and a life expectancy of at least three months. The study design consisted of an initial in-person palliative care consultation, followed by two monthly teleconsultations and a three-month follow-up assessment. Data collection included quantitative analysis from patient-reported questionnaires and qualitative insights from semi-structured interviews. Results: The findings revealed high acceptability, with 100% of recruited patients agreeing to participate in teleconsultation, citing benefits such as more frequent follow-ups and reduced travel-related fatigue. Notably, adherence to teleconsultation was independent of demographic factors, including age, gender, or socioeconomic status. Observed benefits included improved symptom management, enhanced overall quality of care, stronger patient-provider relationships, and a reduction in emergency hospitalizations. However, challenges were identified, including that 64% of patients died within three months, indicating late referral to palliative care, and 5% lacked digital access, highlighting healthcare disparities; additionally, some patients required assistance with technology. Conclusions: Teleconsultation serves as a valuable complementary tool in palliative oncology care, enhancing care continuity and reducing unplanned hospital admissions. Nonetheless, this study underscores the necessity for earlier integration of palliative care and targeted efforts to address digital access inequalities.

- I advise the authors to professionally proofread their manuscript prior to resubmitting.

The manuscript was written by a multidisciplinary team of professionals (physicians, sociologists, ethicists) and reviewed by all authors. Additionally, it has been proofread by a native English speaker.

- Introduction is complet in the health area. However, I think it could better frame the problem under study, indicating more explicitly the factors that were at the basis of the research.

We have added contextual elements to the introduction, and some aspects have also been incorporated at the beginning of the discussion. The key factors underlying our research are:

- An increasing number of patients with advanced-stage cancer but a life expectancy of more than six months will be referred to palliative care departments in the coming years. Advances in cancer treatments allow many patients to transition into a chronic disease phase, requiring follow-up for months or even years without a curative treatment plan. French hospitals are currently considering the implementation of a structured follow-up for these patients, initially in collaboration with oncologists and subsequently under the exclusive care of palliative care teams. The current trend favors the early involvement of palliative care physicians, particularly in gynecologic oncology, to ensure continuity of care when curative treatments are discontinued. Additionally, there is a growing emphasis on deprescription to improve the quality of life in end-of-life patients and to avoid aggressive treatments in the last two months of life, in accordance with ASCO recommendations. These factors will inevitably increase the workload of palliative care teams in the medium term.

- Due to limited human resources and overloaded healthcare structures, hospitals currently offer only quarterly follow-ups for patients exclusively under palliative care. However, rapid health deterioration can occur between two in-person consultations (every three months), often leading to emergency hospitalizations or intensive care admissions, where maximalist interventions are systematically applied. This approach is not consistent with palliative care policies that prioritize quality of life and discourage aggressive treatments in the final month of life.

- It would be beneficial to anticipate these sudden health deteriorations. A monthly follow-up consultation could be a viable solution.

- Patients under palliative care follow-up could benefit from teleconsultations (video consultations).

- Our study aims to assess the acceptability and feasibility of a monthly teleconsultation follow-up for oncology patients, without prior selection. The objective is to determine whether a specific patient profile benefits more from teleconsultations or whether all patients could derive advantages from monthly teleconsultations.

- Methods chapter needs more specification of empirical work. For example, I would like to see a better characterization of the target population and sample in the empirical part

We have added Table 2, which presents the characteristics of each patient individually. For example, we have recorded the patients' primary pathology (prior to referral to the palliative care department), their age at inclusion, the number of teleconsultations performed, their potential participation in a clinical trial before referral, their educational and professional background, whether they live alone, with a partner, or with a parent, their prior experience with teleconsultation, and their place of residence (distance from the healthcare facility).

In structural terms, I suggest that the discussion should be combined with the results and that the conclusions should be in a “CONCLUSIONS” chapter.

The results and discussion have been merged in this revised version of the manuscript, while the conclusion remains separate.

Reviewer #2: The study identifies a predominance of affluent female patients, likely influenced by the institute’s focus on breast cancer. However, it does not adequately explore the implications of this demographic concentration. Provide detailed information on recruitment strategies. Specify whether efforts were made to include a broader demographic, such as patients from varied socioeconomic backgrounds, men, or those with other cancer types. Explicitly discuss how this sample demographic may limit the generalizability of findings, particularly for less affluent, male, or geographically isolated populations. If data permits, include subgroup comparisons (e.g., by gender, socioeconomic status, or geography) to identify variations in teleconsultation experiences or outcomes.

The hospital center where the study was conducted does not exclusively manage patients with breast cancer or, more broadly, gynecological cancers (ovarian, endometrial, cervical). It has a large department dedicated to head and neck cancers, as well as an internationally recognized onco-hematology department specializing in the management of multiple myeloma and lymphomas. Additionally, the center has specialists in brain cancers and melanoma and hosts a major department dedicated to early-phase and phase 3 clinical trials.

The study included 36 patients diagnosed with the following conditions:

• Breast cancer: 11

• Ovarian adenocarcinoma: 6

• Cervical cancer: 2

• Squamous cell carcinoma: 5

• Melanoma: 4

• Rhabdomyosarcoma: 2

• Gastrointestinal stromal tumors: 2

• Lung carcinoma: 3

• Mesothelioma: 1

• Medulloblastoma: 1

Our study aimed to include both male and female patients with various pathologies and from different socio-economic backgrounds.

We have incorporated a new table providing a more detailed presentation of the patient cohort. Additionally, we have elaborated on the hypotheses regarding the underrepresentation of men from disadvantaged socio-economic backgrounds in the manuscript. The key points are as follows:

• Patients with head and neck tumors, who are predominantly male and from lower socio-economic backgrounds, are often put into remission following surgery and are not referred to the palliative care department. In cases of recurrence, physicians do not habitually consider palliative care follow-up.

• Patients with brain tumors frequently have a poor prognosis at diagnosis despite maintaining a good general condition. The initial treatment lines (surgery, chemotherapy, radiotherapy) do not significantly impact their overall health, and they are not referred to palliative care. In cases of recurrence, oncologists typically continue patient management until death.

• In onco-hematology, there is limited integration of palliative care. Patients are referred to palliative care at best within the month preceding death, which is incompatible with monthly follow-up via teleconsultation.

Nevertheless, all included patients completed a written questionnaire on their perceptions and expectations regarding teleconsultation before its implementation. We found no correlation between gender, socio-economic status, pathology, or other factors.

Exclusion of 5% of patients due to lack of internet access underlines inequities in teleconsultation availability. Suggest strategies such as providing technical equipment (e.g., tablets, mobile hotspots); establishing community telemedicine hubs for shared access; and offering telephonic consultations as a low-tech alternative. Discuss how these findings could inform policies aimed at reducing the digital divide, such as public-private partnerships to improve digital infrastructure in underserved areas or subsidies for patients to access necessary technology.

Following this study, we established a partnership with the "French League Against Cancer" (a non-profit organization), which has committed to funding tablets with internet access for palliative care patients in need. In the publication text, we propose this partnership model with patient associations or organizations supporting vulnerable populations (non-governmental organizations).

We have also identified an alternative approach to ensure that individuals without digital tools are not excluded from teleconsultations. In France, pharmacies are widely distributed across the entire territory. The vast majority of French citizens have access to a nearby pharmacy, whether in urban or rural areas. The government has funded the installation of teleconsultation booths in all rural pharmacies and a significant number of urban pharmacies. This initiative provides patients with internet access and a dedicated private space for their consultations. These teleconsultation sessions in pharmacies are offered free of charge.

The study briefly mentions educational, cultural, and medical capital but lacks depth on their influence on teleconsultation outcomes. Incorporate specific patient stories or quotes from interviews that highlight how these forms of capital impact engagement and satisfaction with teleconsultations. Provide actionable insights on how teleconsultation practices can be adapted to different social profiles. For instance, patients with lower digital literacy may benefit from tutorials or simplified interfaces, while those with high medical literacy may prefer in-depth discussions.

As we have demonstrated, there is no correlation between the level of education and patients' acceptance of teleconsultations. All included patients (who were not selected based on this criterion, as all had accepted) successfully connected and benefited from the consultation(s). In the discussion, we highlight the importance of caregivers in facilitating the connection. The possibility for patients to conduct their teleconsultation in a pharmacy, with the assistance of on-site staff and free of charge, suggests that teleconsultations could be accessible to everyone, regardless of their educational background.

A total of 48 consultations were conducted, and physicians did not observe any difference in patient behavior compared to in-person visits. We do not believe that the use of a screen for consultations alters patients' understanding. However, it should be noted that our study was conducted with palliative oncology patients. These patients had been followed in oncology for several months or even years, meaning they had developed a certain level of health literacy.

The high mortality rate (64%) within three months of enrolment indicates delayed referrals to palliative care, limiting teleconsultation benefits. Investigate and discuss factors contributing to late referrals, such as healthcare provider hesitancy, lack of patient awareness, or systemic delays. Advocate for integrating palliative care earlier in the oncology care pathway, emphasising benefits beyond end-of-life care. Reflect on how late-stage enrolment may have constrained patients’ ability to adapt to teleconsultations, potentially skewing satisfaction or utility perceptions.

The reviewer is absolutely right. Patients are referred to the palliative care department far too late. As mentioned earlier, this is partly due to the shortage of palliative care physicians (human resources) but, more importantly, to the lack of awareness among oncologists, surgical oncologists, and onco-hematologists regarding the expertise of palliative care specialists. There is a real need for all practitioners to become more familiar with palliative care.

In the collective imagination (both among patients and healthcare providers), palliative care is often associated with death or, at best, with the final days before passing. The current debates in France regarding the potential legalization of euthanasia only reinforce these preconceptions.

French cancer centers, under the impetus of the National Cancer Institute (INCa), have initiated discussions on the early integration of palliative care into the patient care pathway. Our institution will launch joint consultations involving both palliative care specialists and oncologists starting this fall for patients experiencing a recurrence of gynecological cancer (ovarian or cervical cancer). The primary objectives of these joint consultations are to reduce hospitalizations (we hope that this dual follow-up will lead to fewer hospitalization days for patients in the following year) and, most importantly, to prevent the prescription of overly aggressive treatments at the end of life (deprescription). Additionally, this approach is expected to improve patients' quality of life.

If we can demonstrate that dual follow-up enhances these parameters, such consultations will progressively be implemented within our institution and, more broadly, across the 19 French cancer centers.

However, we do not believe that late recruitment has limited patients' ability to adapt to teleconsultations. All patients have adapted very well, and, as mentioned earlier, they have been followed for several months or even years before their admission to the palliative care department. They are already familiar with the institution and the he

---

## [Decision Letter · Decision Letter 1]

11 Jul 2025

Dear Dr. Couderc,

We look forward to receiving your revised manuscript.

Kind regards,

Alexandre Morais Nunes, Ph.D.

Academic Editor

PLOS ONE

Reviewers' comments:

Reviewer's Responses to Questions

**Comments to the Author**

Reviewer #2: (No Response)

Reviewer #3: (No Response)

2. Is the manuscript technically sound, and do the data support the conclusions?

Reviewer #2: Yes

Reviewer #3: No

3. Has the statistical analysis been performed appropriately and rigorously?

Reviewer #2: N/A

Reviewer #3: N/A

4. Have the authors made all data underlying the findings in their manuscript fully available?

Reviewer #2: Yes

Reviewer #3: No

5. Is the manuscript presented in an intelligible fashion and written in standard English?

Reviewer #2: Yes

Reviewer #3: Yes

Reviewer #2: The study provides rich qualitative insights; it lacks quantitative data that could enhance its generalisability. A mixed-methods approach, incorporating statistical analysis on teleconsultation adherence rates, patient outcomes, and healthcare resource utilisation, would strengthen its findings. Additionally, the study focuses exclusively on women, which limits its applicability to male patients, whose communication styles and healthcare engagement may differ. Future research should explore gendered differences in telemedicine adoption and effectiveness.

The concept of the sentinel patient is useful but may not apply to all palliative care patients. The study predominantly examines patients who are already familiar with medical systems and possess significant health literacy. This overlooks populations with lower medical capital, such as those from disadvantaged socioeconomic backgrounds or with limited prior healthcare engagement. Additionally, while the study suggests that all patients connected without major technical difficulties, it does not explore those who refused teleconsultations or faced digital barriers. This omission risks presenting an overly optimistic view of telemedicine’s accessibility.

The study briefly mentions that older patients had slight hesitations regarding digital tools; it does not provide a more profound analysis of technological barriers. Questions remain regarding i) internet availability in rural areas; ii) the impact of cognitive decline on telemedicine efficacy; and iii) the usability of teleconsultation platforms for those with limited digital literacy. A more detailed discussion of these barriers would provide a more nuanced understanding of telemedicine’s limitations.

The study highlights the convenience of telemedicine; it does not fully address the ethical and emotional implications of conducting palliative care remotely. End-of-life care is deeply personal, and face-to-face interactions often provide comfort to patients and families. The study could have explored i) whether patients felt a diminished sense of empathy or emotional connection in teleconsultations; ii) how physicians adapted their communication styles to convey empathy via screens; and iii) the psychological impact of receiving difficult news remotely.

Reviewer #3: Manuscript Title: The benefits of palliative care follow-up combining day hospital and telemedicine

Thank you for the opportunity to review this revised manuscript. The topic is highly relevant and addresses a pressing issue in modern palliative care, namely how to provide continuity and access amidst limited resources. The hybrid model of day hospital visits supplemented with monthly teleconsultations is well aligned with current clinical trends and the authors have clearly invested significant effort into both the intervention and its reporting.

However, several major concerns remain regarding the scientific structure, methodological transparency, and framing of results. These must be rigorously addressed before, in my opinion, the manuscript can be considered for publication.

1. Structure: Results and Discussion should be separated

The current manuscript blends results and interpretation into a single narrative section. This format is not appropriate for a mixed-methods empirical study. It undermines the clarity and reproducibility of findings and makes it difficult to assess which conclusions are supported by data. The result is a blurring of the line between what was observed and what is interpreted, which compromises clarity, reproducibility, and transparency.

Suggestion: Clearly separate the Results (quantitative and qualitative, with descriptive reporting) from the Discussion (interpretive and reflective). This is essential to align with scientific conventions.

2. Methods section lacks sufficient detail

The manuscript’s methodological reporting is insufficient for both quantitative and qualitative components.

Quantitative: There is no information on item development, scale use, validation, or analytic plan. Statements such as “no correlation was observed” are unsupported.

Qualitative: Thematic analysis is described only briefly. There is no mention of coding procedures, who performed the coding, whether any inter-rater discussion took place, or what software (if any) was used.

Suggestion: Expand the Methods section to provide a clear, replicable account of both arms of data collection and analysis. Avoid making statistical claims unless supported by actual analysis. The Methods section should follow a golden thread that is complemented by the results (i.e. in the same logical order).

3. Title Overstates the Study’s Scope

The current title suggests demonstrated “benefits” from the hybrid care model. In reality, this is a short-term, small-sample feasibility and acceptability study with descriptive data and no comparative outcome evaluation.

Suggestion: Revise the title to more accurately reflect the study’s scope—e.g., “Feasibility of a Hybrid Day Hospital and Teleconsultation Follow-Up Model in Palliative Oncology” or similar.

4. Interpretation should be tempered

The manuscript occasionally draws conclusions not directly supported by the data. For example, claims about symptom management or reduction in emergency admissions are suggestive at best and should be clearly framed as such. Similarly, the interesting “sentinel patient” framing from the sociological literature should be presented more cautiously, given the lack of comparison across patient groups.

Suggestion: Reframe these interpretations as hypotheses or observations for further study, rather than demonstrated effects.

5. Qualitative reporting should be focused

The inclusion of patient quotes is valuable, but currently some quotes are lengthy and repetitive. This can be streamlined to improve readability and thematic clarity.

Suggestion: Shorten representative quotes where possible and consider a summary table of themes and sample quotes for structure and clarity.

6. Generalizability Limitations Need Further Emphasis

The sample is small, skewed toward female and relatively affluent patients and drawn from a single institution. These limitations are acknowledged but should be discussed more directly in relation to the feasibility of broader implementation. The study population is likely to be highly selected (e.g., patients with access to digital tools, caregiver support, good pre-existing health system familiarity).

Conclusion

This is a valuable and well-motivated pilot study and I appreciate the authors’ clear dedication to improving care delivery. However, substantial revisions are required to meet basic standards of scientific reporting. I recommend major revision, with clear restructuring of the manuscript and strengthening of methods and framing. I would be willing to re-review a revised version that addresses these concerns.

**Do you want your identity to be public for this peer review?** For information about this choice, including consent withdrawal, please see our Privacy Policy

Reviewer #2: **Yes:** Ricardo de Moraes e Soares

Reviewer #3: **Yes:** Patrick Altmann, MD

---

## [Author Response · Author response to Decision Letter 2]

1 Oct 2025

Point-by-point responses to the referees

We thank the referees for the time they devoted to evaluating our manuscript. We are grateful for their constructive feedback, which has allowed us to improve both the presentation and the discussion of our findings.

Below you will find our detailed, point-by-point responses to the reviewers’ comments.

Referee 1

The study provides rich qualitative perspectives; however, it lacks quantitative data that could strengthen its generalizability. A mixed-methods approach including statistical analysis of teleconsultation adherence rates, patient outcomes, and healthcare resource utilization would reinforce the conclusions.

We acknowledge the many biases in our study. We have expanded the Methods section so that the referees can better assess how we generated the quantitative analyses. The study was conducted in France, where the healthcare system provides free access to care for all. Consequently, adherence to a monthly rather than quarterly follow-up device is 100%. The contribution of teleconsultations to patient medical follow-up is difficult to quantify in a study such as ours with a small sample size. Physicians reported 4 cases out of 38 where teleconsultation provided significant added value (these data are included in the revised manuscript). We have shown that monthly follow-up allows more reactive adjustments in management than if the patient had access to a palliative care physician only every 3 months. We also demonstrate that monthly follow-up improves patients’ perceived quality of life—they feel reassured by monthly specialist follow-up—though we have not quantified this effect.

Furthermore, the study focuses exclusively on women, which limits applicability to male patients, whose communication styles and engagement in care may differ. Future research should explore gender differences in telemedicine adoption and effectiveness.

We fully agree with the referee’s observation. Please find our remarks regarding this study bias included in the Discussion. In this study, we included patients without prior selection, based on admission to the department. We included both men and women with various pathologies and from diverse socio-economic backgrounds. Patients with head and neck tumors (predominantly men, often from less favorable socio-economic contexts) are underrepresented because they are frequently put into remission after surgery and rarely referred to the palliative care mobile team. In cases of recurrence, physicians in the department often do not think to propose palliative care follow-up. Patients with brain tumors often have a poor prognosis at diagnosis despite a good general condition. First-line treatments (surgery, chemotherapy, radiotherapy) do not alter their general condition and they are not referred to palliative care. In case of recurrence, oncologists follow these patients until death with only occasional input from palliative care physicians. Finally, in hemato-oncology, there is limited integration of palliative care; patients are referred at best within the month preceding death, making monthly teleconsultation follow-up unfeasible.

The concept of a “sentinel patient” is relevant but may not apply to all palliative care patients. The study mainly examines patients already familiar with healthcare systems and possessing good health literacy, thereby neglecting populations with lower medical capital, such as those from disadvantaged backgrounds or with limited prior contact with healthcare.

Our study concerned patients who had been followed for their disease for many years. It is true that they all had good health literacy. We believe that teleconsultations would not be suitable for patients with late-stage disease detected too late (without any specific treatment available), those in precarious situations, or those with a short life expectancy. Teleconsultations are, in our view, best suited for the long-term follow-up of patients already well known to the care teams.

Moreover, while the study indicates that all patients connected without major technical issues, it does not address those who refused teleconsultations or faced digital barriers. This omission risks providing an overly optimistic view of telemedicine accessibility.

We have added in the Discussion the possibility for patients to use teleconsultation platforms available in pharmacies (well-established throughout France). These platforms are freely accessible and allow all patients, regardless of social level, to benefit from teleconsultations. They offer broad opening hours nationwide. A study on the use of these platforms in palliative care could be considered.

The study briefly mentions that older patients showed some hesitation in using digital tools; however, it does not provide an in-depth analysis of technological barriers. Questions remain regarding:

i) Internet availability in rural areas;

ii) The impact of cognitive decline on telemedicine effectiveness;

iii) Platform ergonomics for those with low digital literacy.

We suggested the use of pharmacy teleconsultation platforms, particularly in rural areas.

Regarding cognitive decline, we believe that teleconsultations are not suitable without a caregiver’s presence.

Pharmacy platforms provide assistance with connecting and, if patients wish, pharmacists or pharmacy assistants can help with understanding the physician’s communication.

The study highlights the convenience of telemedicine but does not fully address the ethical and emotional implications of remote palliative care. End-of-life care is deeply personal, and face-to-face interactions often provide comfort. The study could have explored:

i) whether patients felt a loss of empathy or emotional connection;

ii) how physicians adapted their communication style to convey empathy through a screen;

iii) the psychological impact of delivering bad news remotely.

Patients assessed via telemedicine mostly wished to continue such follow-up beyond the study—most alternated in-person and teleconsultation visits, while 2 chose teleconsultations only.

Emotional moments occurred during teleconsultations, sometimes involving both patient and caregiver, often around care and support, loss of autonomy, and approaching death.

Empathy via telemedicine requires time; only vision and hearing are available channels, as touch is not possible. In practice, however, not all patients rely on physical contact for communication.

Regarding breaking bad news, telephone is indeed a poor tool due to the inability to gauge non-verbal reactions; teleconsultations, like in-person visits, allow access to non-verbal cues (only touch is missing, see above).

Reviewer #3 Comments:

Title of manuscript: “Benefits of palliative care follow-up combining day hospital and telemedicine.”

We appreciate the opportunity to review this revised manuscript. The topic is highly relevant and addresses an important issue in palliative care: ensuring continuity and accessibility of care despite limited resources. The hybrid model combining day-hospital visits and monthly teleconsultations is aligned with current clinical trends, and the authors have clearly invested significant effort in the intervention and its presentation.

However, major concerns remain regarding scientific structure, methodological transparency, and interpretation of results, which should be rigorously addressed before publication.

1. Structure: Results and Discussion must be separated.

We have followed the referee’s recommendation by restructuring the manuscript. The Results section is now more factual, while interpretation and reflection are presented separately in the Discussion. A brief concluding section remains. A tracked-changes version is provided for review.

2. Methods section lacks detail.

The methodological description was previously insufficient for both the quantitative and qualitative components.

Quantitative: No information was provided regarding item development, the scales used, their validation, or the analysis plan. Statements such as “no correlation observed” were not substantiated.

Qualitative: The thematic analysis was described only briefly, with no details on coding procedures, the individuals involved in coding, whether inter-rater discussions took place, or whether any software was used.

Suggestion (as per reviewer): The Methods section should be expanded to offer a clear and reproducible description of both data collection and analysis procedures. Statistical claims should be supported by actual analyses. The Methods should follow a logical structure reflected in the Results.

We apologize to the referee for the lack of detail in the previous version. We have now carefully expanded the description of the methods, which we hope are now sufficiently clear. The questionnaire and the interview guide translated into English are provided in Annexes 1 and 2.

Data Analysis

Quantitative Analyses:

Upon inclusion in the study, each patient was assigned an identification number linked to the paper questionnaire they completed. This identification number enabled tracking of the patient’s trajectory (e.g., number of teleconsultations performed) and, when applicable, linking the transcript of the follow-up interview conducted at 3 months. Only the principal investigator (VMS) and the research coordinator (BC) had access to the tables linking patient identities to research-related data.

Questionnaires were analyzed using Excel. The objective was to capture patients’ initial apprehension or confidence toward teleconsultations as well as their knowledge and understanding of the system. Responses were compared according to patient characteristics (age, place of residence, education level, occupational category, etc.). We also compared pre-teleconsultation questionnaire responses with data transcribed from the semi-structured interview for 11 patients. The data were subjected to simple statistical analyses (frequency and mean) and cross-tabulations.

Raw questionnaire data were additionally coded using IRAMUTEQ lexical analysis software (IRAMUTEQ R 3.1.2; free version of ALCESTE) [24]. For each included patient, we recorded gender, age at inclusion, occupational status, education level, place of residence, and the presence of a caregiver. We also extracted certain questionnaire responses (e.g., apprehension or interest regarding teleconsultations [Y/N], need for caregiver assistance to connect [Y/N], etc.). Finally, we added the number of teleconsultations performed and, when available, the corpus of the final interview (IRAMUTEQ allows a distributional statistical analysis to provide significant word occurrences within a corpus).

Item-by-item correlation analyses were performed using Jamovi software [25], which processes IRAMUTEQ outputs. These analyses were verified by the team epidemiologist (SL). Correlation analyses regarding favorable or unfavorable a priori attitudes toward teleconsultations were carried out by combining responses to favorable items (1, 2, 3, 4, 10, 11, 13, and 15) and unfavorable items (5, 6, 7, 8, 9, 12, 14, and 16).

Qualitative Analyses:

We conducted a sociological thematic analysis of the 11 interviews [26], as well as an IRAMUTEQ-based lexical analysis. The IRAMUTEQ analysis of interview transcripts did not identify any significant text clusters. Regarding the sociological thematic analysis, the method used was manual and theme-based, consisting of transposing the corpus into a set of representative themes related to the research question. The corpus was analyzed by the research team, which included a sociologist (AFR), an ethicist (BC), the head of the institute’s palliative care department (NCH), and the physicians (VMS, MB). The thematic sociological analysis was produced through the triangulated perspectives of the study authors. The thematic analysis thus combined the quantitative outputs from IRAMUTEQ/Jamovi with an interpretive analysis of the meaning conveyed in participants’ responses. No additional software was used.

3. The title overstates the scope of the study :The current title suggests that “benefits” have been demonstrated as a result of the hybrid model. In reality, this is a short-term feasibility and acceptability study, based on a small sample and descriptive data, without any comparative evaluation of outcomes.

Suggestion: Revise the title to better reflect the actual scope of the study, for example: “Feasibility of a hybrid follow-up model combining day-hospital care and teleconsultation in palliative oncology” or an equivalent formulation.

4.

We propose a new title consistent with the reviewer’s suggestion: “Palliative care follow-up for cancer patients combining day-hospital visits and telemedicine: what feasibility?”

4. Interpretation must be more nuanced

The manuscript sometimes draws conclusions that are not directly supported by the data. For example, statements regarding symptom management or the reduction of emergency admissions remain hypothetical and should be presented as such. Similarly, the interesting sociological concept of the “sentinel patient” should be articulated with greater caution, given the absence of comparisons between patient groups.

Suggestion: Rephrase these interpretations as hypotheses or avenues for future research, rather than demonstrated effects.

We are fully aware that our small sample does not allow for generalization, and we have revised the discussion accordingly. We have also provided a more detailed description of four patients who clearly derived measurable benefit from teleconsultations in the Results section:

“Some teleconsultations allowed the improvement of distressing symptoms: for two patients who experienced increased pain, management consisted of an adjustment of analgesic treatment for one and the initiation of a WHO step III analgesic for the other. The advantage of these two teleconsultations compared to a simple telephone call was that they allowed us to observe how the patient moved and provided information relevant to the etiological diagnosis (despite the absence of a physical examination). A prescription was then sent to the pharmacy and to the patient.

For another patient, in light of a deteriorating general condition and difficulties maintaining care at home, the initiation of hospital-at-home (HAD) services was proposed: this patient was able to die at home without transfer to the emergency department or inpatient ward (contrary to her wishes). For another patient, a systematic teleconsultation allowed the identification of a femoral fracture: she underwent plain radiography confirming the fracture; her case was then directly discussed in the pain and bone metastasis multidisciplinary tumor board, and she was admitted directly to orthopedic surgery, bypassing the emergency department, which would have been ill-suited in this palliative context. Finally, for another patient, visualization of an exudative tumor wound during the teleconsultation, in collaboration with the home-care nurse present, allowed adaptation of the care protocol and thus improved the patient’s comfort.

In the absence of these tailored interventions or treatment modifications, these patients would likely have required hospitalization, or even ended up in the emergency department—settings often inappropriate in the palliative context and not necessarily aligned with patient wishes.

Moreover, in oncology, due to the long medical trajectory, patients often “wait” until the situation has severely deteriorated before seeking care through emergency services, even when this is not optimal. It should be noted that for the patients described above in our study, the systematic use of teleconsultations helped identify symptoms, even though patients had not alerted the mobile palliative care team despite having their contact information. Each issue was addressed and contributed to improving their quality of life. However, we acknowledge that, given our small sample size, these findings can only be considered exploratory and represent directions for future research.”

5. The qualitative report should be more focused

The inclusion of patient quotations is valuable; however, some excerpts are lengthy and redundant, which undermines the readability and clarity of the thematic

---

## [Decision Letter · Decision Letter 2]

12 Jan 2026

PONE-D-25-00301R2Palliative care follow-up for cancer patients combining day-hospital visits and telemedicine: what feasibility?PLOS One?

Dear Dr.  Couderc,

We look forward to receiving your revised manuscript.

Kind regards,

Alexandre Morais Nunes, Ph.D.

Academic Editor

PLOS One

Journal Requirements:

Additional Editor Comments:

The revision turned out much better. While I believe reviewer 4's suggestion of a major revision is excessive, I ask that you address the methodological issues that the reviewer identified.

Reviewer's Responses to Questions

**Comments to the Author**

Reviewer #1: All comments have been addressed

Reviewer #4: (No Response)

2. Is the manuscript technically sound, and do the data support the conclusions?

Reviewer #1: Yes

Reviewer #4: No

3. Has the statistical analysis been performed appropriately and rigorously?

Reviewer #1: Yes

Reviewer #4: No

4. Have the authors made all data underlying the findings in their manuscript fully available?

Reviewer #1: Yes

Reviewer #4: No

5. Is the manuscript presented in an intelligible fashion and written in standard English?

Reviewer #1: Yes

Reviewer #4: Yes

Reviewer #1: The authors have made all the suggested revisions. It is ready to be approved.

I have nothing to add. I think it is aligned with the journal rules.

Reviewer #4: The methodology is described in a valid manner. The text states that a feasibility assessment model was adopted. However, the adopted model was not aligned with the assumptions for conducting feasibility studies. The text is silent regarding the justification of the recruitment rate, retention parameters, acceptance thresholds, and feasibility criteria. The feasibility analysis is retrospective and descriptive. The situation limits the validity of the analysis and hinders the evaluation of the model's application. It is suggested that the feasibility outcomes be defined a priori, such as the completion rates of admissible teleconsultations, the proportion of eligible patients, and the acceptable rates of technical failures. It is also suggested that inferences regarding feasibility be related to predefined benchmarks, and in case of impossibility, the limitations should be acknowledged and that feasibility was assessed in an exploratory rather than systematic manner.

The inclusion criterion, estimated life expectancy ≥3 months, is not operationalised or justified, and the imprecision is evidenced by the high mortality rate (64%). The situation raises concerns about the validity of the selection and adequacy of the eligibility assessment, which affects the interpretation of the results. It is suggested that the clinical criteria for estimating life expectancy be described. It is also suggested to discuss the implications of inaccurate prognostication on the validity of the study. The limitations and biases should be discussed, and the sample reformulated.

Quantitative methods depend on specific items from the questionnaire, which have not been validated or subjected to psychometric evaluations. The text is also silent regarding the reliability, construct validity, or consistency of the composite indices. The conclusions about the patients' behaviours, reassurance, and perceived benefits are based on measures of uncertain validity. It is suggested to describe the assumptions behind the construction of the questionnaire or, alternatively, to acknowledge the lack of validation, which represents a significant methodological limitation. It is also suggested to exercise more moderation in interpreting the questionnaire results and that the conclusions be limited solely to the observed evidence.

The statistical analyses are insufficient to support the inferences of absence of associations between demographic variables and the feasibility or acceptability of teleconsultation. Due to the sample size, skewed distributions, and limited statistical power, non-significant results cannot be interpreted as evidence of the absence of effect. It is suggested to reformulate the text. It is also suggested to replace definitive terms (did not influence) with more probabilistic terms (no statistically detected association in the sample), acknowledging the risk of Type II error. The text should avoid conclusions related to equity that are not supported by the obtained results. Qualitative analysis lacks methodological rigour to support conceptual interpretations.

The description of the results suggests the existence of beneficial contributions from teleconsultation. However, the data only allow us to state that teleconsultation enabled clinical interventions in individual cases but do not allow us to conclude that it caused improvements in clinical outcomes or that it prevented hospitalisations. Given the absence of comparison terms, the interpretations of the results exceed the evidentiary strength of the data. It is suggested to reclassify the examples as illustrations of descriptive cases rather than observed evidence. The interpretations have a hypothetical nature and are speculative.

There is an observed confusion between feasibility, acceptability, and perceived benefit throughout the Results and Discussion sections. Although patient satisfaction and positive perceptions have been documented, the results do not equate to feasibility in the target population, especially considering the exclusion of patients without digital access and those who passed away early. It is suggested that feasibility (implementation capacity), acceptability (patient perception), and perceived benefit (subjective value) be described as distinct constructs with their respective limitations separated. It is also suggested that evidence be emphasised.

**Do you want your identity to be public for this peer review?** For information about this choice, including consent withdrawal, please see our Privacy Policy

Reviewer #1: **Yes:** Andreia Matos

Reviewer #4: No

---

## [Author Response · Author response to Decision Letter 3]

14 Feb 2026

Referee 4

The methodology is described in a valid manner. The text states that a feasibility assessment model was adopted. However, the adopted model was not aligned with the assumptions for conducting feasibility studies. The text is silent regarding the justification of the recruitment rate, retention parameters, acceptance thresholds, and feasibility criteria. The feasibility analysis is retrospective and descriptive. The situation limits the validity of the analysis and hinders the evaluation of the model's application. It is suggested that the feasibility outcomes be defined a priori, such as the completion rates of admissible teleconsultations, the proportion of eligible patients, and the acceptable rates of technical failures. It is also suggested that inferences regarding feasibility be related to predefined benchmarks, and in case of impossibility, the limitations should be acknowledged and that feasibility was assessed in an exploratory rather than systematic manner.

To evaluate the feasibility of teleconsultations among all patients with access to a connected device, we deliberately applied no additional selection criteria beyond a WHO performance status ≥ 3. Age, accompaniment status, and educational level were not considered exclusion criteria.

We have added two sentences to the Methods section clarifying how acceptability and feasibility were assessed. Acceptability corresponded to the percentage of patients with connected devices who agreed to participate in the protocol when it was proposed to them (100%). Feasibility corresponded to the percentage of patients who were able to successfully connect to their medical consultations (48 teleconsultations completed out of 48 scheduled, including two that were postponed by a few days due to holidays or poor internet connectivity).

Our aim was to demonstrate that teleconsultation could be successfully implemented regardless of patients’ age or digital literacy. All included patients were able to connect and complete their teleconsultation, supporting the feasibility of this approach provided that access to a connected device is ensured.

In the Discussion section, we outline potential solutions for patients lacking access to connected devices. Given the reliability of the internet infrastructure in France, major technical failures were not anticipated. In cases of network or device malfunction, appointments could be rescheduled, as would be done for in-person visits. This occurred in 2 of the 48 consultations conducted.

The inclusion criterion, estimated life expectancy ≥3 months, is not operationalised or justified, and the imprecision is evidenced by the high mortality rate (64%). The situation raises concerns about the validity of the selection and adequacy of the eligibility assessment, which affects the interpretation of the results. It is suggested that the clinical criteria for estimating life expectancy be described. It is also suggested to discuss the implications of inaccurate prognostication on the validity of the study. The limitations and biases should be discussed, and the sample reformulated.

To date, medicine does not provide reliable scientific tools to accurately quantify the number of weeks or months of remaining life expectancy. Moreover, in this palliative and symptom-oriented context, and considering the benefit–risk balance within an ethical framework, we chose not to perform additional biological investigations and did not use biological parameters as selection criteria.

Inclusion in the study was based on clinical criteria only: absence of major vital organ failure (hepatic, pulmonary, or cerebral involvement), a WHO performance status ≤ 3, and the clinical judgment of the disease-specific oncologist referring patients to the palliative care department, who estimated a life expectancy greater than three months.

These characteristics have been added to the Materials and Methods section.

The primary objective of this exploratory study was to assess the feasibility and acceptability, among patients receiving palliative cancer care, of monthly remote follow-up (teleconsultation) with a physician they had met only once (i.e., without an established therapeutic relationship), with the aim of implementing this model sustainably in comprehensive cancer centers.

The potential clinical impact in terms of preventing emergency department admissions through monthly rather than quarterly follow-up is mentioned in the manuscript as an observational finding only. Given the small sample size, no statistically robust conclusion could be drawn. This was not the primary objective of the study, as no longitudinal clinical outcome data were collected during follow-up. The manuscript has been revised (abstract and main text) to clarify that clinical benefit was not the primary endpoint, particularly due to the absence of predefined evaluation criteria.

Although a high mortality rate was observed, this does not affect the validity of our feasibility study, which was not designed as a quantitative outcome study. Mortality limited the number of final interviews that could be conducted. Nevertheless, the total number of teleconsultations performed (n = 48) and the number of completed final interviews (n = 11) consistently support the feasibility of teleconsultation, even in a palliative oncology setting.

Teleconsultation was perceived as beneficial by patients, who reported that monthly follow-up was more appropriate and reassuring than a single quarterly in-person visit. The qualitative interviews were consistent, and data saturation was achieved.

To evaluate feasibility without focusing on a specific patient subgroup, participants were included consecutively as they were referred to the department. Patients with prior contact with the service (i.e., with an existing physician–patient relationship) were excluded in order to avoid bias related to a pre-established therapeutic bond.

Quantitative methods depend on specific items from the questionnaire, which have not been validated or subjected to psychometric evaluations. The text is also silent regarding the reliability, construct validity, or consistency of the composite indices.

Participants completed a self-administered questionnaire. The instrument included various types of items: single- and multiple-choice questions, as well as 5-point Likert-type scales. Likert scales were used to assess the degree of agreement with specific statements (e.g., ranging from “strongly disagree” to “strongly agree”). These formats were selected to better capture patients’ opinions, perceptions, and preferences, while remaining brief and easy to complete for patients who were often fatigued.

The survey focused on several key domains:

• General perceptions of telemedicine (feasibility)

• Perceived personal relevance and adequacy of telemedicine within their individual care trajectory

• Relationships with medical specialists

• Relationships with relatives

These domains were intended to explore the human and social dimensions of patients’ perspectives on this model of follow-up.

We have elaborated on this point in the Materials and Methods section, as suggested by the reviewer.

The questionnaire was developed collaboratively by all co-authors of the publication (physicians, a sociologist, statisticians, and ethicists). It was reviewed and approved by the Research Ethics Committee of the Federal University, which includes four psychologists among its members. The committee formally accredited the study and validated the methodological framework as part of its area of expertise (IRB number assigned).

The conclusions about the patients' behaviours, reassurance, and perceived benefits are based on measures of uncertain validity. It is suggested to describe the assumptions behind the construction of the questionnaire or, alternatively, to acknowledge the lack of validation, which represents a significant methodological limitation. It is also suggested to exercise more moderation in interpreting the questionnaire results and that the conclusions be limited solely to the observed evidence.

We have revised the manuscript to adopt a more cautious tone, as suggested by the reviewer. Specifically, we have moderated our statements by clearly indicating that the study is qualitative, descriptive rather than aiming at providing a quantitative evaluation. This study is exploratory, descriptive, and qualitative, and was not designed to provide quantitative conclusions.

The statistical analyses are insufficient to support the inferences of absence of associations between demographic variables and the feasibility or acceptability of teleconsultation. Due to the sample size, skewed distributions, and limited statistical power, non-significant results cannot be interpreted as evidence of the absence of effect.

We did not state that there was no effect but only that “No correlation was observed between genre, residence, education or profession and attitudes teleconsultation”. We completely agree with the reviewer on the fact that sample size did not allow for inferences and that is why we did not adopt a quantitative reading of the results. As we stated in the discussion, in the recruitment and sample characteristics section “This study included a small sample of 36 patients. Therefore, the quantitative descriptive analyses should only be considered as illustrative of the results of the qualitative analyses, which were the core of this work.”

It is suggested to reformulate the text. It is also suggested to replace definitive terms (did not influence) with more probabilistic terms (no statistically detected association in the sample), acknowledging the risk of Type II error. The text should avoid conclusions related to equity that are not supported by the obtained results. Qualitative analysis lacks methodological rigour to support conceptual interpretations.

We have revised the manuscript in accordance with the reviewer’s comments. We have also removed the conclusion regarding equity from the abstract, as suggested.

The description of the results suggests the existence of beneficial contributions from teleconsultation. However, the data only allow us to state that teleconsultation enabled clinical interventions in individual cases but do not allow us to conclude that it caused improvements in clinical outcomes or that it prevented hospitalizations. Given the absence of comparison terms, the interpretations of the results exceed the evidentiary strength of the data. It is suggested to reclassify the examples as illustrations of descriptive cases rather than observed evidence. The interpretations have a hypothetical nature and are speculative.

We agree with the reviewer and, as mentioned above, have revised the manuscript accordingly.

There is an observed confusion between feasibility, acceptability, and perceived benefit throughout the Results and Discussion sections. Although patient satisfaction and positive perceptions have been documented, the results do not equate to feasibility in the target population, especially considering the exclusion of patients without digital access and those who passed away early. It is suggested that feasibility (implementation capacity), acceptability (patient perception), and perceived benefit (subjective value) be described as distinct constructs with their respective limitations separated. It is also suggested that evidence be emphasized.

We have reorganized the manuscript by separating the sections addressing feasibility, acceptability, and benefits, in accordance with the reviewer’s recommendations.

We are pleased to submit a revised version incorporating a comprehensive set of corrections addressing the comments from both reviewers. Please find enclosed the final version as well as a tracked-changes version to facilitate your assessment of the revisions.

We sincerely hope that our responses meet the reviewers’ expectations.

---

## [Editor Report · Decision Letter 3]

22 Feb 2026

Palliative care follow-up for cancer patients combining day-hospital visits and telemedicine: what feasibility?

PONE-D-25-00301R3

Dear Dr. Bettina C Couderc

We’re pleased to inform you that your manuscript has been judged scientifically suitable for publication and will be formally accepted for publication once it meets all outstanding technical requirements.

Kind regards,

Alexandre Morais Nunes, Ph.D.

Academic Editor

PLOS One

Additional Editor Comments (optional):

The authors submitted a second revised version, incorporating a comprehensive set of corrections that address the comments from both reviewers. In addition to attaching the new final version, they took the trouble to include a version with tracked changes to facilitate the evaluation of the revisions. As the editor, I compared the two versions and believe that after long and demanding revisions, the paper is ready to be approved.

Reviewers' comments:

In the previous review, Reviewer 1 had already accepted the work, albeit with minor suggestions. Reviewer 2 demanded more changes (even somewhat excessively), which the authors accepted.

---

## [Editor Report · Acceptance letter]

PONE-D-25-00301R3

PLOS One

Dear Dr. Couderc,

I'm pleased to inform you that your manuscript has been deemed suitable for publication in PLOS One. Congratulations! Your manuscript is now being handed over to our production team.

Kind regards,

on behalf of

Professor Alexandre Morais Nunes

Academic Editor

PLOS One